# Dendritic trafficking faces physiologically critical speed-precision tradeoffs

Alex H Williams[1,2,3*], Cian O'Donnell[2,4], Terrence J Sejnowski[2,5], Timothy O'Leary[6,7*]

[1]Department of Neurosciences, University of California, San Diego, La Jolla, United States; [2]Howard Hughes Medical Institute, Salk Institute for Biological Studies, La Jolla, United States; [3]Department of Neurobiology, Stanford University, Stanford, United States; [4]Department of Computer Science, University of Bristol, Bristol, United Kingdom; [5]Division of Biological Sciences, University of California, San Diego, La Jolla, United States; [6]Volen Center and Biology Department, Brandeis University, Waltham, United States; [7]Department of Engineering, University of Cambridge, Cambridge, United Kingdom

**Abstract** Nervous system function requires intracellular transport of channels, receptors, mRNAs, and other cargo throughout complex neuronal morphologies. Local signals such as synaptic input can regulate cargo trafficking, motivating the leading conceptual model of neuron-wide transport, sometimes called the 'sushi-belt model' (Doyle and Kiebler, 2011). Current theories and experiments are based on this model, yet its predictions are not rigorously understood. We formalized the sushi belt model mathematically, and show that it can achieve arbitrarily complex spatial distributions of cargo in reconstructed morphologies. However, the model also predicts an unavoidable, morphology dependent tradeoff between speed, precision and metabolic efficiency of cargo transport. With experimental estimates of trafficking kinetics, the model predicts delays of many hours or days for modestly accurate and efficient cargo delivery throughout a dendritic tree. These findings challenge current understanding of the efficacy of nucleus-to-synapse trafficking and may explain the prevalence of local biosynthesis in neurons.

*For correspondence: ahwillia@ stanford.edu (AHW); timothy. oleary@eng.cam.ac.uk (TO)

**Competing interests:** The authors declare that no competing interests exist.

## Introduction

Dendritic and axonal trees of neurons often have many tens or even thousands of branches that can extend across the entire nervous system. Distributing biomolecular cargo within neuronal morphologies is therefore a considerable logistical task, especially for components that are synthesized in locations distant from their site of use. Nonetheless, molecular transport is important for many neurophysiological processes, such as synaptic plasticity, neurite development and metabolism. For example, long-lasting forms of synaptic plasticity appear to depend on anterograde transport of mRNAs (*Nguyen et al., 1994*; *Bading, 2000*; *Kandel, 2001*) and specific mRNAs are known to be selectively transported to regions of heightened synaptic activity (*Steward et al., 1998*; *Steward and Worley, 2001*; *Moga et al., 2004*) and to developing synaptic contacts (*Lyles et al., 2006*).

On the other hand, local biosynthesis and component recycling are known to support dendritic physiology, including some forms of synaptic plasticity (*Kang and Schuman, 1996*; *Aakalu et al., 2001*; *Vickers et al., 2005*; *Sutton and Schuman, 2006*; *Holt and Schuman, 2013*) and maintenance of cytoskeletal, membrane and signalling pathways (*Park et al., 2004, 2006*; *Grant and Donaldson, 2009*; *Zheng et al., 2015*). Neurons therefore rely on a mixture of local metabolism and global transport, but the relative contributions of these mechanisms are not understood. Analyzing

**eLife digest** Neurons are the workhorses of the nervous system, forming intricate networks to store, process and exchange information. They often connect to many thousands of other cells via intricate branched structures called neurites. This gives neurons their complex tree-like shape, which distinguishes them from many other kinds of cell.

However, like all cells, neurons must continually repair and replace their internal components as they become damaged. Neurons also need to be able to produce new components at particular times, for example, when they establish new connections and form memories. But how do neurons ensure that these components are delivered to the right place at the right time? In some cases neurons simply recycle components or make new ones where they are needed, but experiments suggest that they transport other essential components up and down neurites as though on a conveyor belt. Individual parts of a neuron are believed to select certain components they need from those that pass by. But can this system, which is known as the sushi-belt model, distribute material to all parts of neurons despite their complex shapes?

Using computational and mathematical modeling, Williams et al. show that this model can indeed account for transport within neurons, but that it also predicts certain tradeoffs. To maintain accurate delivery, neurons must be able to tolerate delays of hours to days for components to be distributed. Neurons can reduce these delays, for example, by manufacturing more components than they need. However, such solutions are costly. Tradeoffs between the speed, accuracy and efficiency of delivery thus limit the ability of neurons to adapt and repair themselves, and may constrain the speed and accuracy with which they can form new connections and memories.

In the future, experimental work should reveal whether the relationships predicted by this model apply in real cells. In particular, studies should examine whether neurons with different shapes and roles fine-tune the delivery system to suit their particular needs. For example, some neurons may tolerate long delays to ensure components are delivered to the exactly the right locations, while others may prioritize speedy delivery.

the performance of global trafficking provides a principled way to understand the division of labor between local and global mechanisms.

In this paper, we examine how well trafficking can perform given what we know about active transport and the typical morphologies of neurites. There are two parts to this question. First, how can active transport achieve specific spatial distributions of cargo using only local signals? Second, how long does it take to distribute cargo to a given degree of accuracy and what factors contribute to delays?

Intracellular trafficking is being characterized in increasing detail (*Buxbaum et al., 2014b*; *Hancock, 2014*; *Wu et al., 2016*). Microscopic cargo movements are stochastic, bidirectional, and inhomogeneous along neurites, leading to to the hypothesis that trafficking is predominantly controlled by local pathways that signal demand for nearby cargo, rather than a centralized addressing system (*Welte, 2004*; *Bressloff and Newby, 2009*; *Newby and Bressloff, 2010a*; *Doyle and Kiebler, 2011*; *Buxbaum et al., 2015*). These local signals are not fully characterized, but there is evidence for multiple mechanisms including transient elevations in second-messengers like $Ca^{2+}$ and ADP (*Mironov, 2007*; *Wang and Schwarz, 2009*), glutamate receptor activation (*Kao et al., 2010*; *Buxbaum et al., 2014b*), and changes in microtubule-associated proteins (*Soundararajan and Bullock, 2014*).

A leading conceptual model ties together these details by proposing that local signalling and regulation of bidirectional trafficking determines the spatial distribution of cargo in neurons (*Welte, 2004*; *Buxbaum et al., 2015*). *Doyle and Kiebler (2011)* call this the 'sushi belt model'. In this analogy, molecular cargoes are represented by sushi plates that move along a conveyor belt, as in certain restaurants. Customers sitting alongside the belt correspond to locations along a dendrite that have specific and potentially time-critical demand for the amount and type of sushi they consume, but they can only choose from nearby plates as they pass.

Stated in words, the sushi belt model is an intuitive, plausible account of the molecular basis of cargo distribution. Yet it is unclear whether this model conforms to intuition, and whether it implies unanticipated predictions. Can this trafficking system accurately generate global distributions of cargo using only local signals? Does the model predict cross-talk, or interference between spatially separated regions of the neuron that require the same kind of cargo? How quickly and how accurately can cargo be delivered by this model, given what is known about trafficking kinetics, and do these measures of performance depend on morphology or the spatial pattern of demand?

We address these questions using simple mathematical models that capture experimentally measured features of trafficking. We confirm that the sushi-belt model can produce any spatial distribution of cargo in complex morphologies. However, the model also predicts that global trafficking from the soma is severely limited by tradeoffs between the speed, efficiency, robustness, and accuracy of cargo delivery. Versions of the model predict testable interactions between trafficking-dependent processes, while the model as a whole suggests that time-critical processes like synaptic plasticity may be less precise, or less dependent on global transport than is currently assumed.

## Results

### A simple model captures bulk behaviour of actively transported cargo

Transport along microtubules is mediated by kinesin and dynein motors that mediate anterograde and retrograde transport, respectively (*Block et al., 1990*; *Hirokawa et al., 2010*; *Gagnon and Mowry, 2011*). Cargo is often simultaneously bound to both forms of motor protein, resulting in stochastic back-and-forth movements with a net direction determined by the balance of opposing movements (*Welte, 2004*; *Hancock, 2014*; *Buxbaum et al., 2014a*, *Figure 1A*). We modelled this process as a biased random walk, which is general enough to accommodate variations in biophysical details (*Bressloff, 2006*; *Bressloff and Earnshaw, 2007*; *Müller et al., 2008*; *Bressloff and Newby, 2009*; *Newby and Bressloff, 2010a*; *Bressloff and Newby, 2013*).

*Figure 1* shows this model in a one-dimensional cable, corresponding to a section of neurite. In each unit of time the cargo moves a unit distance forwards or backwards, or remains in the same place, each with different probabilities. In the simplest version of the model, the probabilities of forward and backward jumps are constant for each time step (*Figure 1B*, top panel). Cargo can also undergo extended unidirectional runs (*Klumpp and Lipowsky, 2005*; *Müller et al., 2008*; *Hancock, 2014*). The model can account for these runs with jump probabilities that depend on the previous movement of the particle (*Figure 1B*, bottom panel, Materials and methods).

While the movement of individual cargoes is stochastic, the spatial distribution of a population (*Figure 1C*) changes predictably. This is seen in *Figure 1D*, which shows the distribution of 1000 molecules over time, without (top panel) and with (bottom panel) unidirectional runs. The bulk distribution of cargo can therefore be modelled as a deterministic process that describes how cargo concentration spreads out in time.

A convenient and flexible formulation of this process is a mass-action model (*Voit et al., 2015*) that spatially discretizes the neuron into small compartments. In an unbranched neurite with $N$ compartments, the mass-action model is:

$$u_1 \underset{b_1}{\overset{a_1}{\rightleftharpoons}} u_2 \underset{b_2}{\overset{a_2}{\rightleftharpoons}} u_3 \underset{b_3}{\overset{a_3}{\rightleftharpoons}} \dots \underset{b_{N-1}}{\overset{a_{N-1}}{\rightleftharpoons}} u_N \tag{1}$$

where $u_i$ is the amount of cargo in each compartment, and $a_i$ and $b_i$ denote trafficking rate constants of cargo exchange between adjacent compartments. This model maps onto the well-known drift-diffusion equation when the trafficking rates are spatially homogeneous (*Figure 1E*; *Smith and Simmons, 2001*). We used this to constrain trafficking rate constants based on single-particle tracking experiments (*Dynes and Steward, 2007*) or estimates of the mean and variance of particle positions from imaging experiments (*Roy et al., 2012*, see Materials and methods).

With a compartment length of 1 μm, the simulations in *Figure 1D* gave mean particle velocities of 15 μm per minute, which is within the range of experimental observations for microtubule transport (*Rogers and Gelfand, 1998*; *Dynes and Steward, 2007*; *Müller et al., 2008*). The variances of the particle distributions depended on whether unidirectional runs are assumed, and respectively grew at a rate of ~0.58 and ~1.33 μm$^2$ per second for the top and bottom panels of *Figure 1D*. The

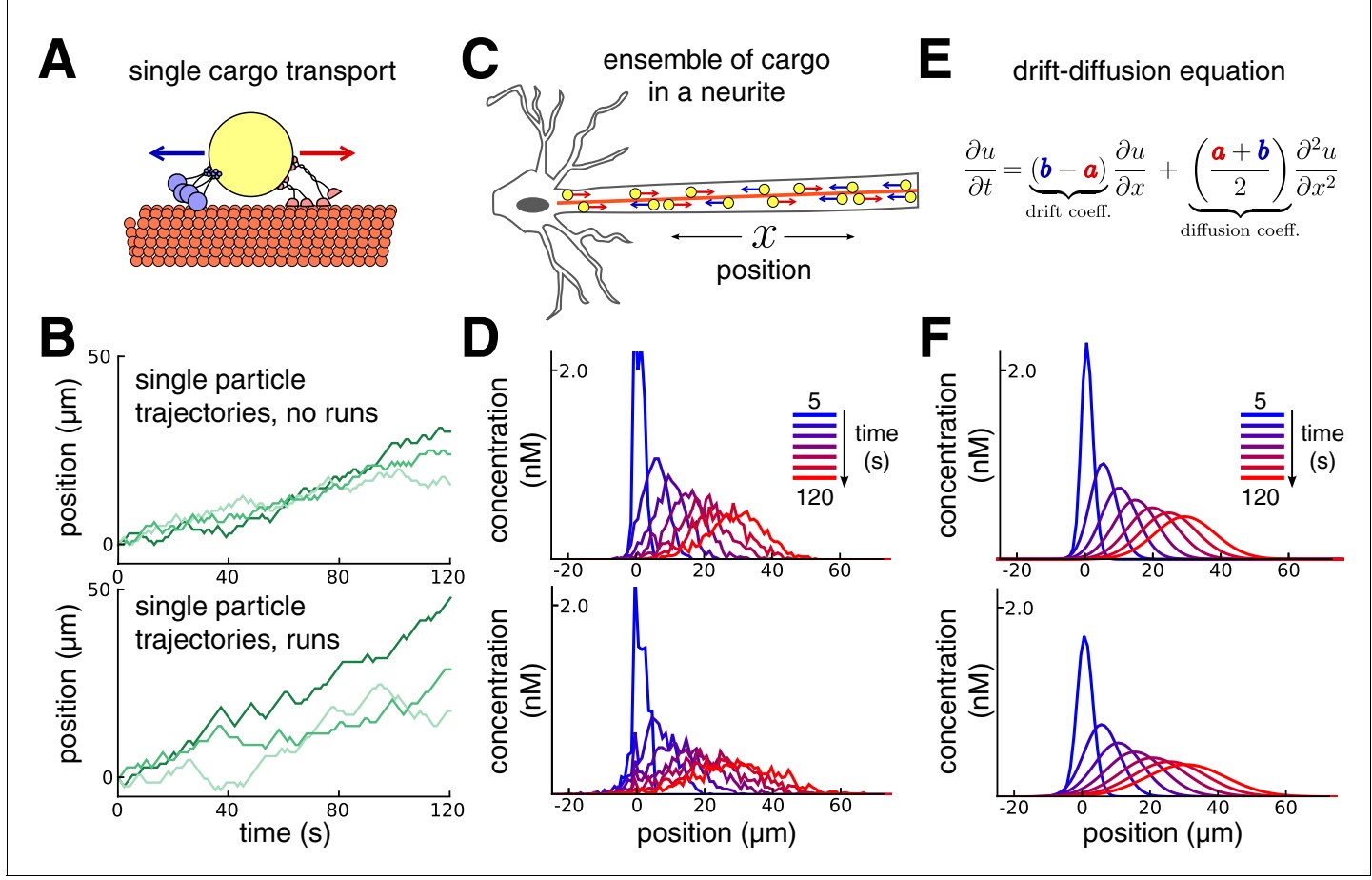

**Figure 1.** Constructing a coarse-grained model of intracellular transport. (A) Cartoon of a single cargo particle on a microtubule attached to opposing motor proteins. (B) Three example biased random walks, representing the stochastic movements of individual cargoes. (Top panel) A simple random walk with each step independent of previous steps. (Bottom panel) Adding history-dependence to the biased random walk results in sustained unidirectional runs and stalls in movement. (C) Cartoon of a population of cargo particles being transported along the length of a neurite. (D) Concentration profile of a population of cargoes, simulated as 1000 independent random walks along a cable/neurite. (Top panel) simulations without runs. (Bottom panel) Simulations with runs. (E) In the limit of many individual cargo particles, the concentration of particles $u$ is described by a drift diffusion model whose parameters, $a$ and $b$, map onto the mass action model (*Equation 1*). (F) The mass-action model provides a good fit to the simulations of bulk cargo movement in (D). (Top panel) Fitted trafficking rates for the model with no runs were $a \approx 0.42 \text{ s}^{-1}$, $b \approx 0.17 \text{ s}^{-1}$. (Bottom panel) Fitting the model with runs gives $a \approx 0.79 \text{ s}^{-1}$, $b \approx 0.54 \text{ s}^{-1}$.

The following figure supplement is available for figure 1:

**Figure supplement 1.** The effect of cargo run length on mass-action model fit and diffusion coefficient.

mass action model provides a good fit to both cases (*Figure 1F*). In general, the apparent diffusion coefficient of the model increases as run length increases (*Figure 1—figure supplement 1A*). The accuracy of the mass-action model decreases as the run length increases. However, the model remains a reasonable approximation for many physiological run lengths and particle numbers, even over a relatively short time window of 100 s (*Figure 1—figure supplement 1B*).

## Biophysical formulation of the sushi belt model

The advantage of the mass action model is that it easily extends to complex morphologies with spatially non-uniform trafficking rates, and can accommodate additional processes, including sequestration of cargo. The sushi-belt model (*Doyle and Kiebler, 2011*) proposes that local mechanisms modify local trafficking rates and capture cargo as it passes. For these local signals to encode the demand for cargo, some feedback mechanism must exist between the local concentration of cargo

and the signal itself. There are many biologically plausible mechanisms for locally encoding demand (see Materials and methods). For our main results, we did not focus on these details and simply assumed a perfect demand signal. We have thus addressed the performance of the transport mechanism per se, with the most forgiving assumptions about the reliability of the demand signal.

The mass action model of sushi-belt transport is:

$$u_1 \underset{b_1}{\overset{a_1}{\rightleftharpoons}} u_2 \underset{b_2}{\overset{a_2}{\rightleftharpoons}} u_3 \underset{b_3}{\overset{a_3}{\rightleftharpoons}} u_4 \underset{b_4}{\overset{a_4}{\rightleftharpoons}} \dots$$

$$c_1 \downarrow \qquad c_2 \downarrow \qquad c_3 \downarrow \qquad c_4 \downarrow$$

$$u_1^\star \qquad u_2^\star \qquad u_3^\star \qquad u_4^\star \tag{2}$$

where $u$ represents the concentration of cargo on the network of microtubules, indexed by the compartment. In each compartment, molecules can irreversibly detach from the microtubules in a reaction $u_i \overset{c_i}{\rightarrow} u_i^\star$, where $u_i^\star$ denotes the detached cargo. Biologically, cargo will eventually degrade. However, in this study we are concerned with how cargo can be rapidly distributed so that detached cargo can satisfy demand for at least some time. Therefore, for simplicity we assume degradation rates are effectively zero.

We first asked whether modifying the trafficking rates alone was sufficient to reliably distribute cargo. Thus, we set all detachment rate constants ($c_i$) to zero, and considered a model with trafficking only between compartments, as shown in *Figure 2A*. Mathematical analysis shows that, for a fixed set of trafficking parameters, the distribution of cargo approaches a unique steady-state distribution over time, regardless of the initial distribution of cargo. The steady-state occurs when the ratio of cargo concentrations between neighboring compartments is balanced by the trafficking rates:

$$\left. \frac{u_p}{u_c} \right|_{ss} = \frac{b}{a} \tag{3}$$

where $u_p$ is the level in a 'parent' compartment (closer to soma), $u_c$ is the level in the adjacent 'child' compartment (closer to periphery) and $b$ and $a$ are the trafficking rate constants between these compartments.

If $\tilde{u}_i$ represents the local demand signal in compartment $i$, then *Equation (3)* gives the condition for cargo distribution to match demand:

$$\frac{b}{a} = \frac{\tilde{u}_p}{\tilde{u}_c} \tag{4}$$

An example demand profile and the corresponding trafficking rate relationships are shown in *Figure 2B*. This condition ensures that cargo is delivered in proportion to local demand. The absolute concentration at steady-state is determined by the total amount of cargo produced (*Figure 2—figure supplement 1*); in the case of mRNA, this might be controlled at the somatic compartment by transcriptional regulation. In this paper, we focus on the relative accuracy of cargo distribution when some fixed amount of cargo is produced at the soma.

To illustrate demand-modulated trafficking in a realistic setting, we used a reconstructed model of a CA1 pyramidal neuron (*Migliore and Migliore, 2012*). To provide a demand signal, we modelled excitatory synaptic input at 120 locations within three dendritic regions (red dots, *Figure 2C*) and set demand, ($\tilde{u}_i$), equal to the average membrane potential in each electrical compartment (see Materials and methods). As expected, cargo was transported selectively to regions of high synaptic activity (*Video 1*), matching the demand profile exactly at steady state (*Figure 2D*). Therefore, local control of trafficking rates (equivalently, motor protein kinetics) can deliver cargo to match arbitrarily complex spatial demand.

## Transport bottlenecks occur when trafficking rates are non-uniform

We next investigated the consequences of solely modifying trafficking rates to distribute cargo. A particularly striking prediction of this model is that changes in trafficking (or, equivalently, demand signals) in regions close to the soma can strongly affect cargo delivery times to distal sites. As the

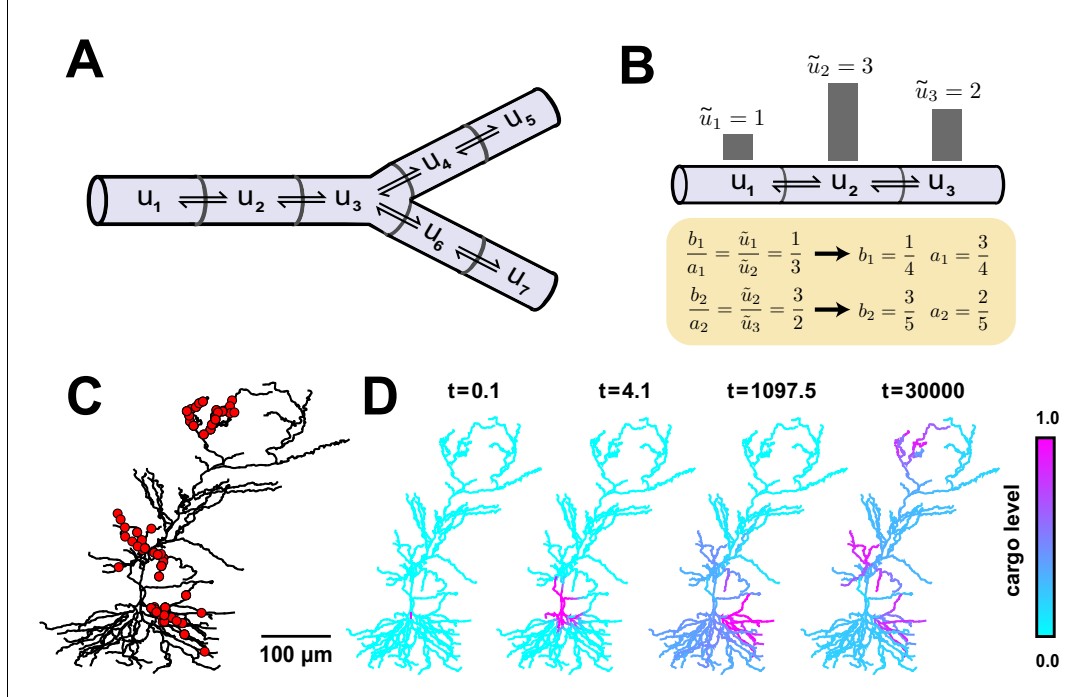

**Figure 2.** Local trafficking rates determine the spatial distribution of biomolecules by a simple kinetic relationship. (A) The mass action transport model for a simple branched morphology. (B) Demonstration of how trafficking rates can be tuned to distribute cargo to match a demand signal. Each pair of rate constants ($\{a_1, b_1\}$, $\{a_2, b_2\}$) was constrained to sum to one. This constraint, combined with the condition in *Equation (4)*, specifies a unique solution to achieve the demand profile. (C) A model of a CA1 pyramidal cell with 742 compartments adapted from (*Migliore and Migliore, 2012*). Spatial cargo demand was set proportional to the average membrane potential due to excitatory synaptic input applied at the locations marked by red dots. (D) Convergence of the cargo concentration in the CA1 model over time, *t* (arbitrary units).

The following figure supplement is available for figure 2:

**Figure supplement 1.** *Equation 4* specifies the relative distribution of cargo, changing the total amount of cargo scales this distribution.

demand signal $\tilde{u}_i$ approaches zero in a compartment, the trafficking rates into that compartment also approach zero, cutting off the flow of cargo along the neurite (*Figure 3A*). The smallest demand signal, $\epsilon$, often determines the rate-limiting time constant for cargo delivery to an entire dendritic tree. We refer to this scenario as a 'transport bottleneck.' *Figure 3A–C* illustrate how decreasing $\epsilon$ to zero causes arbitrarily slow delivery of cargo in a simple three-compartment model.

To illustrate bottlenecks in a more realistic setting, we imposed a bottleneck in the reconstructed CA1 model by setting demand in the middle third of the apical dendrite to a lower level than the rest of the dendritic tree, which was set uniformly high. As expected, the cargo distribution converged much more quickly for uniform demand than with a bottleneck present (*Figure 3D*).

However, less intuitive effects are seen on the convergence times of cargo in specific compartments. *Figure 3E* plots convergence time for $u_i$ to reach a fraction of the steady state value for each compartment. While distal compartments showed prolonged convergence times, (*Figure 3E*, upper right portion of plot), the bottleneck shortened the transport delay to proximal compartments (*Figure 3E*, lower left portion of plot). This occurs because the bottleneck decreases the effective size of proximal part the neuron: cargo spreads efficiently throughout the proximal dendrites, but traverses the bottleneck more slowly.

Another counterintuitive effect is seen when demand varies independently at proximal and distal locations, as might occur during selective synaptic stimulation (see e.g., *Han and Heinemann, 2013*). In *Figure 3F* we simulated demand at proximal and distal portions of the apical dendrite independently and quantified the total convergence time. Proximal demand alone (*Figure 3F* 'proximal') resulted in the fastest convergence time. Convergence was slowest when the demand was

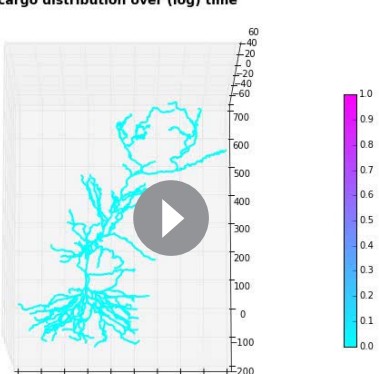

**cargo distribution over (log) time**

**Video 1.** Distribution of trafficked cargo over logarithmically spaced time points in a CA1 pyramidal cell model adapted from (*Migliore and Migliore, 2012*). Cargo was trafficked according to *Equation 4* to match a demand signal established by stimulated synaptic inputs (see *Figure 2C*). Time and cargo concentrations are reported in arbitrary units.

restricted to distal dendrites (*Figure 3F*, 'distal'). Interestingly, when both distal and proximal sites signalled demand (*Figure 3F* 'both'), convergence was substantially faster than the distal-only case, even though cargo still needed to reach the distal neurites. Uniform demand across the entire tree (*Figure 3F* 'entire cell') resulted in a similarly short convergence time.

Together, these results show that locally modulating trafficking movements will have testable effects on global transport times. The presence and relative contribution of this mechanism can be probed experimentally by characterizing the convergence rate of a cargo that aggregates at recently activated synapses, such as *Arc* mRNA (*Steward et al., 1998*). This could be achieved using quantitative optical measurements in combination with synaptic stimulation at specific synaptic inputs.

## Local control of trafficking and detachment results in a family of trafficking strategies

We next considered the full sushi-belt model (*Equation 2*) with local demand signals controlling both trafficking and detachment rate constants (*Figure 4A*). This provides additional flexibility in how cargo can be distributed, since the model can distribute cargo by locally modulating trafficking rates, detachment rates, or both (*Figure 4B*). If trafficking is much faster than detachment ($a, b \gg c$), then the previous results (*Figures 2–3*) remain relevant since the distribution of cargo on the microtubules will approach a quasi-steady state described by *equation (3)*; cargo may then detach at a slow, nonspecific rate ($c_i =$ constant, with $c \ll a, b$). *Figure 4C* shows an example of this scenario, which we call demand-dependent trafficking (DDT). The spatial distribution of cargo is first achieved along the microtubules (red line, *Figure 4C*), and maintained as cargo detaches (blue line, *Figure 4C*).

Alternatively, models can match demand by modulating the detachment process rather than microtuble trafficking. In this case, the trafficking rates are spatially uniform ($a_i = b_i$) so that cargo spreads evenly, and the detachment rates are set proportionally to the local demand, $\tilde{u}_i^{\star}$:

$$c_i \propto \frac{\tilde{u}_i^{\star}}{\tilde{u}_i} \tag{5}$$

The result of this strategy, which we call demand-dependent detachment (DDD), is shown in *Figure 4D*. Unlike DDT, DDD avoids the transport bottlenecks examined in *Figure 3*, and can achieve target patterns with $\tilde{u}_i^{\star}$ equal to zero in certain compartments by setting $c_i = 0$.

Mixed strategies that locally modulate both detachment and trafficking are also able to deliver cargo to match demand. *Figure 4E* shows the behavior of a model whose parameters are a linear interpolation between pure DDT and DDD (see Materials and methods).

## Rapid cargo delivery in the sushi-belt model is error-prone

Although it is mathematically convenient to separate the timescales of trafficking and detachment in the model, this separation may not exist in biological systems tuned for rapid transport. However, removal of timescale separation in the sushi-belt model results in mistargeted delivery of cargo, as we now show.

We returned to the CA1 model of *Figures 2–4* and considered a scenario where there is demand for cargo at the distal apical dendrites (*Figure 5A*). If the detachment rate constants are sufficiently

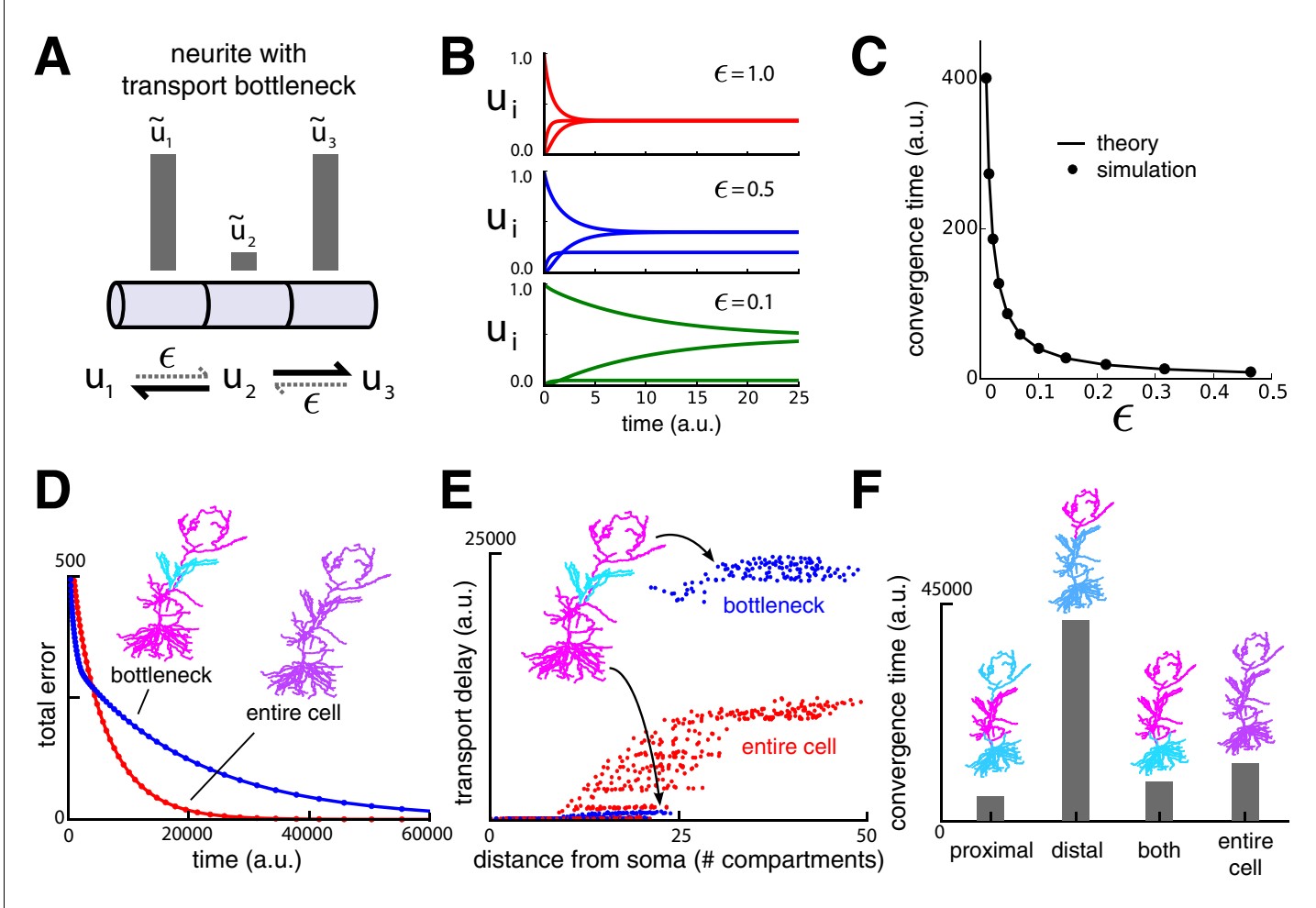

**Figure 3.** Transport bottlenecks caused by cargo demand profiles. (A) A three-compartment transport model, with the middle compartment generating a bottleneck. The vertical bars represent the desired steady-state concentration of cargo in each compartment. The rate of transport into the middle compartment is small ($\epsilon$, dashed arrows) relative to transport out of the middle compartment. (B) Convergence of cargo concentration in all compartments of model in (A) for decreasing relative bottleneck flow rate, $\epsilon$. (C) Simulations (black dots) confirm that the time to convergence is given by the smallest non-zero eigenvalue of the system (solid curve). (D) Convergence to a uniform demand distribution (red line) is faster than a target distribution containing a bottleneck (blue line) in the CA1 model. Total error is the sum of the absolute difference in concentration from demand ($L_1$ norm). Neuron morphologies are color-coded according to steady state cargo concentration. (E) Transport delay for each compartment in the CA1 model (time to accumulate 0.001 units of cargo). (F) Bar plot comparison of the convergence times for different spatial demand distributions in the CA1 model (steady-state indicated in color plots). The timescale for all simulations in the CA1 model was normalized by setting $a_i + b_i = 1$ for each compartment.

slow, then, as before, delivered cargo matched demand nearly exactly in both the DDT and DDD models (*Figure 5A*, left). Increasing detachment rates led to faster convergence, but resulted in cargo leaking off the microtubule on the way to its destination (*Figure 5A*, right). Thus, for a fixed trafficking timescale, there is a tradeoff between the speed and accuracy of cargo delivery. The tradeoff curve shown in *Figure 5B* shows that both accuracy and convergence time decreased smoothly as the detachment rates were increased. This tradeoff was present regardless of whether the trafficking rates (*Figure 5B*, red line) or detachment rates (*Figure 5B*, blue line) were modified to meet demand (compare to *Figure 4C and D*, respectively). However, DDD outperformed DDT in this scenario, since the latter caused bottlenecks in proximal dendrites.

We considered a second scenario in which there was a uniform distribution of demand throughout the entire apical tree (*Figure 5C*). As before, fast detachment led to errors for both transport

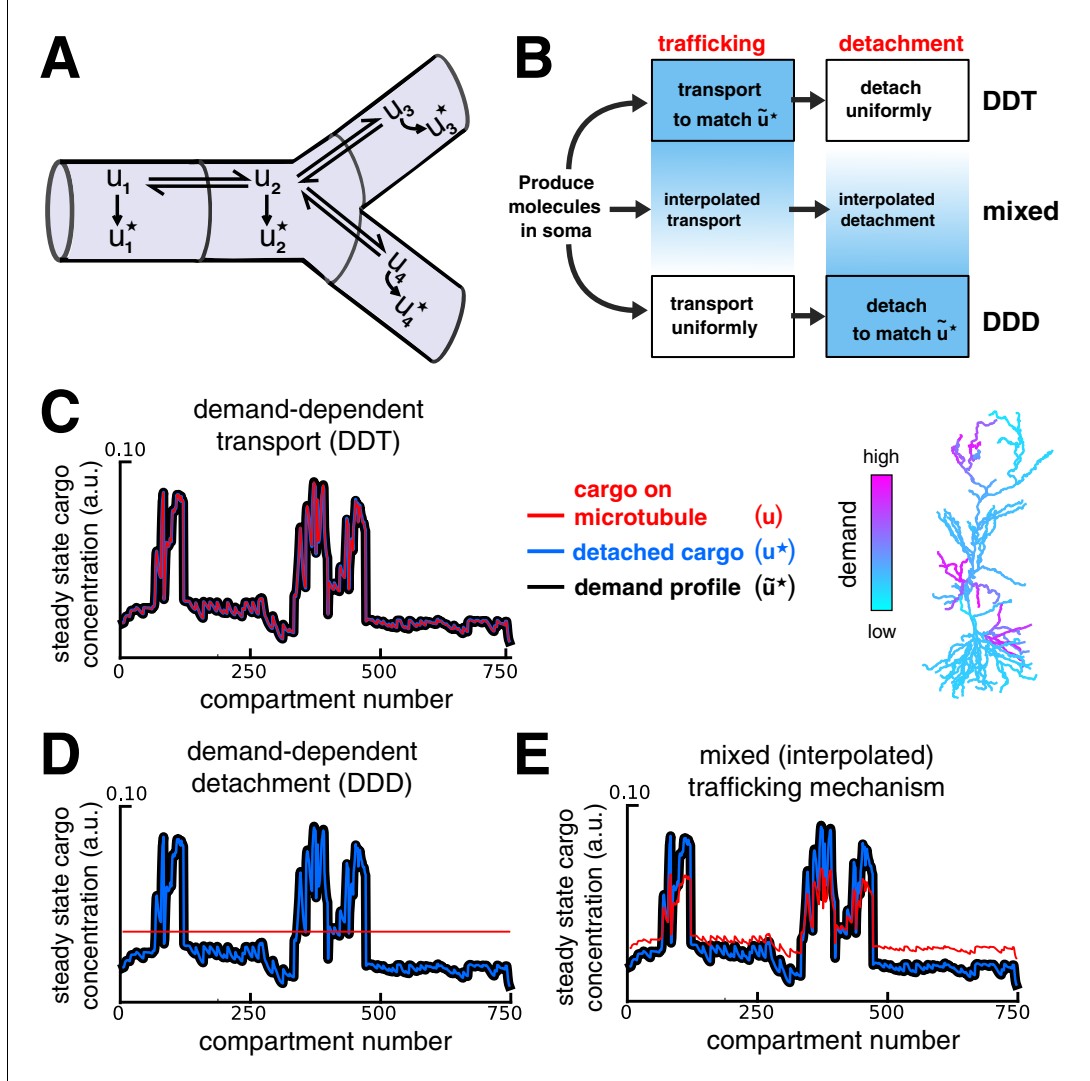

**Figure 4.** Multiple strategies for transport with trafficking and cargo detachment controlled by local signals. (A) Schematic of microtubular transport model with irreversible detachment in a branched morphology. (B) Multiple strategies for trafficking cargo to match local demand (demand = $\tilde{u}^\star$). (Top) The demand-dependent trafficking mechanism (DDT). When the timescale of detachment is sufficiently slow, the distribution of cargo on the microtubules approaches a quasi-steady-state that matches $\tilde{u}^\star$ spatially. This distribution is then transformed into the distribution of detached cargo, $u^\star$. (Bottom) The demand dependent detachment (DDD) mechanism. Uniform trafficking spreads cargo throughout the dendrites, then demand is matched by slowly detaching cargo according to the local demand signal. An entire family of mixed strategies is achieved by interpolating between DDT and DDD. (C–E) Quasi-steady-state distribution of cargo on the microtubules ($u$, red) and steady-state distribution of detached cargo ($u^\star$, blue), shown with a demand profile ($\tilde{u}^\star$, black) for the various strategies diagrammed in panel B. The demand profile is shown spatially in the color-coded CA1 neuron in the right of panel C. Detached cargo matches demand in all cases.

strategies, this time by occluding cargo delivery to distal synaptic sites (*Figure 5C*, right). A smooth tradeoff between speed and accuracy was again present, but, in contrast to *Figure 5A–B*, the DDT model outperformed DDD (*Figure 5D*). Intuitively, DDT is better in this case because DDD results in cargo being needlessly trafficked to the basal dendrites.

Together, these results show that increasing the speed of cargo delivery comes at the cost of accuracy, and that the performance of different trafficking strategies depends on the spatial profile of demand. The balance between demand-dependent trafficking and detachment could be probed experimentally. For example, one could perform an experiment in which distal and proximal synaptic pathways are stimulated independently, while optically monitoring the trafficking of proteins and

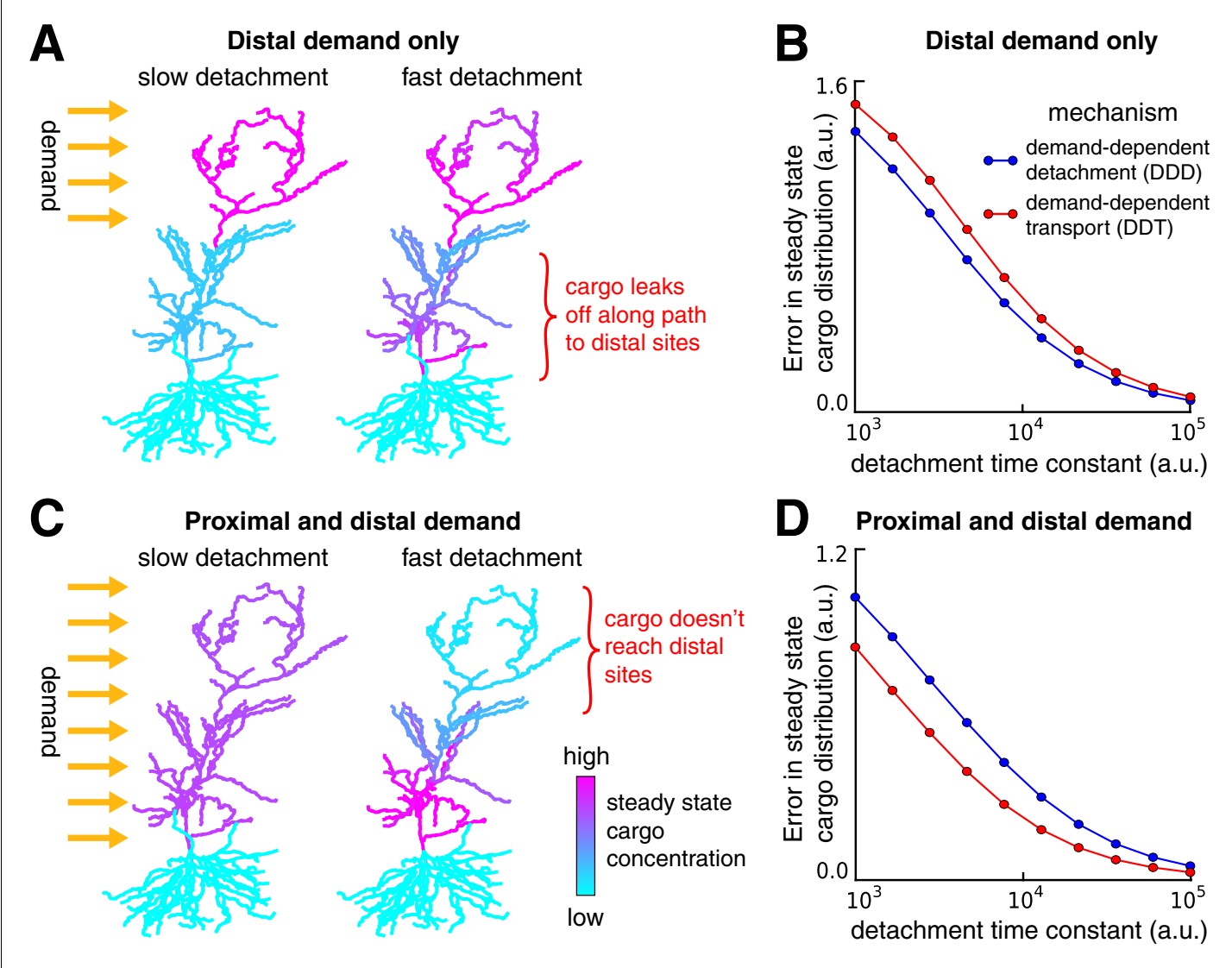

**Figure 5.** Tradeoffs in the performance of trafficking strategies depends on the spatial pattern of demand. (**A**) Delivery of cargo to the distal dendrites with slow (left) and fast detachment rates (right) in a reconstructed CA1 neuron. The achieved pattern does not match the target distribution when detachment is fast, since some cargo is erroneously delivered to proximal sites. (**B**) Tradeoff curves between spatial delivery error and convergence rate for the DDT (red line, see **Figure 4C**) and DDD (blue line, see **Figure 4D**) trafficking strategies. (**C–D**) Same as (**A–B**) but with uniform demand throughout proximal and distal locations. The timescale of all simulations was set by imposing the constraint that $a_i + b_i = 1$ for each compartment, to permit comparison.

mRNAs that are known to be selectively distributed at recently activated synapses. Interactions of the kind seen in **Figure 5A,C** and **Figure 3F** would allow one to infer whether DDT, DDD or a mixture of both strategies are implemented biologically.

## Fine-tuned trafficking rates and cargo recycling introduce new tradeoffs

We next wanted to understand (a) how severe the speed-accuracy tradeoff might be, given experimental estimates of neuron size and trafficking kinetics, and (b) whether simple modifications to the sushi-belt model could circumvent this tradeoff. We examined the DDD model in an unbranched cable with a realistic neurite length (800 μm) and an optimistic diffusion coefficient of 10 μm² s⁻¹, which we set by inversely scaling the trafficking rate constants with the squared compartment length

(see Materials and methods and *Figure 6—figure supplement 1*). All cargo began in the leftmost compartment and was delivered to a small number of demand 'hotspots' (black arrows, *Figure 6A*). Similar results were found when the DDT model was examined in this setting (data not shown).

When the detachment timescale was sufficiently slow, the cargo was distributed evenly across the demand hotspots, even when the spatial distribution of the hotspots was changed (*Figure 6A1*; *Video 2*). Increasing the detachment rate caused faster convergence, but erroneous delivery of cargo. In all cases, hotspots closer to the soma received disproportionate high levels of cargo (*Figure 6A2*; *Video 3*). Importantly, the tradeoff between these extreme cases was severe: it took over a day to deliver 95% of cargo with 10% average error, and over a week to achieve 1% average error (blue line, *Figure 6B*).

We next attempted to circumvent this tradeoff by two strategies. First, motivated by the observation that too much cargo was delivered to proximal sites in *Figure 6A2*, we increased the

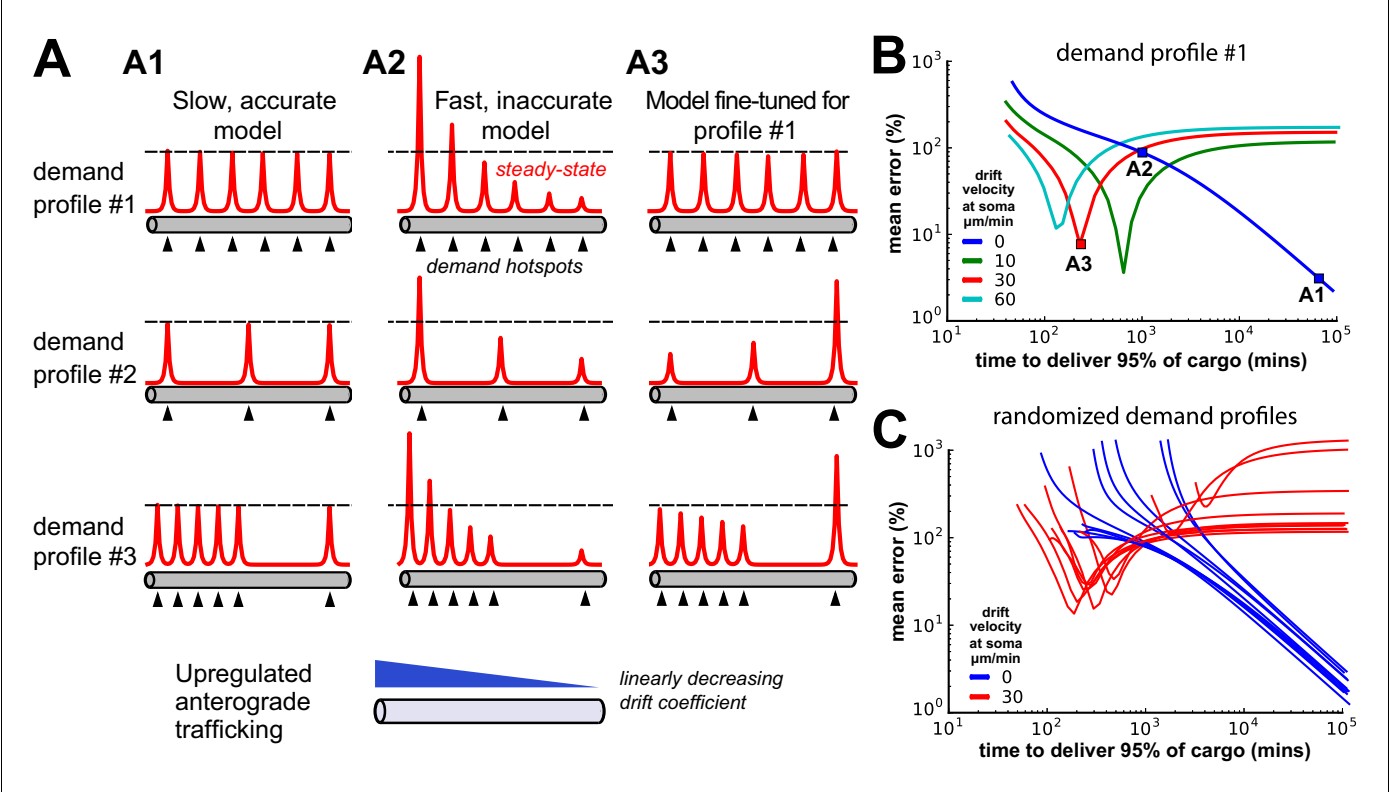

**Figure 6.** Tuning the DDD model for speed and specificity results in sensitivity to the target spatial distribution of cargo. (**A**) Cargo begins on the left end of an unbranched cable to be distributed equally amongst several demand 'hotspots'. Steady-state cargo profiles (red) are shown for three different models (A1, A2, A3) and three different spatial patterns of demand (rows). The bottom panel shows an upregulated anterograde trafficking profile introduced to reduce delivery time in A3; soma is at the leftmost point of the cable. (**A1**) A model with sufficiently slow detachment achieves near-perfect cargo delivery for all demand patterns. (**A2**) Making detachment faster produces quicker convergence, but errors in cargo distribution. (**A3**) Transport rate constants, $a_i$ and $b_i$, were tuned to optimize the distribution of cargo for the first demand pattern (top row); detachment rate constants were the same as in model A2. (**B**) Tradeoff curves between non-specificity and convergence rate for six evenly spaced demand hotspots (the top row of panel A). Tradeoff curves are shown for the DDD model (blue line) as well as models that combine DDD with the upregulated anterograde trafficking profile (as in A, bottom panel). Marked points denote where models A1, A2, A3 sit on these tradeoff curves. (**C**) Tradeoff curves for randomized demand patterns (six uniformly placed hotspots). Ten simulations are shown for the DDD model with (red) and without (blue) anterograde trafficking upregulation.

The following figure supplement is available for figure 6:

**Figure supplement 1.** Changing compartment size over an order of magnitude leads to insignificant changes in model behavior when trafficking rates are appropriately scaled (i.e. $a_i$ and $b_i$ are inversely scaled to the squared compartment length; the diffusion coefficient converges to 10 $\mu m^2 s^{-1}$ as the compartment size shrinks to zero).

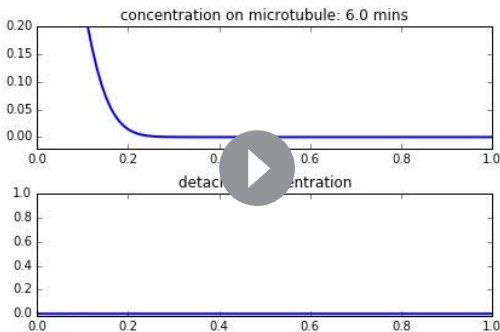 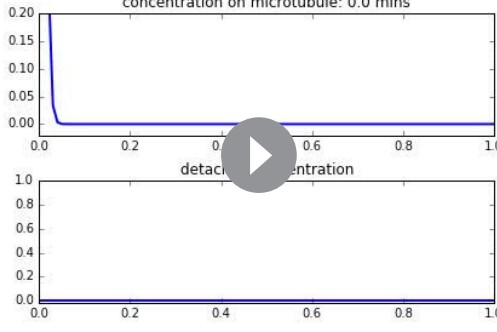

**Video 2.** A model with slow detachment rate accurately distributes cargo to six demand hotspots in an unbranched cable. The spatial distribution of detached cargo (bottom subplot) and cargo on the microtubules (top subplot) are shown over logarithmically spaced timepoints. Compare to **Figure 6A1** (top row).

**Video 3.** A model with a fast detachment rate misallocates cargo to six demand hotspots in an unbranched cable. The spatial distribution of detached cargo (bottom subplot) and cargo on the microtubules (top subplot) are shown over logarithmically spaced timepoints. Proximal demand hotspots receive too much cargo, while distal regions receive too little. Compare to **Figure 6A2** (top row).

anterograde trafficking rate of cargo near the soma so that more cargo would reach distal sites. By carefully fine-tuning a linearly decreasing profile of trafficking bias (illustrated in **Figure 6A**, bottom panel), we obtained a model (**Figure 6A3**; **Video 4**) that provided accurate and fast delivery (within 10% error in 200 min) for a distribution of six, evenly placed hotspots.

However, this model's performance was very sensitive to changes in the spatial pattern of demand (**Figure 6A3**, middle and bottom; **Video 5**). Increasing the anterograde trafficking rates produced nonmonotonic speed-accuracy tradeoff curves (green, red, and cyan curves **Figure 6B**), indicating that the detachment rates needed to be fine-tuned to produce low error. Randomly altering the spatial profile of demand hotspots resulted in variable tradeoff curves for a fine-tuned trafficking model (red lines, **Figure 6C**); an untuned model was able to achieve more reliable cargo delivery albeit at the cost of much slower delivery times (blue lines, **Figure 6C**).

Next, we considered a variant of the sushi-belt model that allowed for the reversible detachment/ reattachment of cargo from the microtubules (**Figure 7A**):

$$
\begin{array}{ccccccccc}
u_1 & \overset{a_1}{\underset{b_1}{\rightleftharpoons}} & u_2 & \overset{a_2}{\underset{b_2}{\rightleftharpoons}} & u_3 & \overset{a_3}{\underset{b_3}{\rightleftharpoons}} & u_4 & \overset{a_4}{\underset{b_4}{\rightleftharpoons}} & \cdots \\[4pt]
d_1 \Big\updownarrow c_1 & & d_2 \Big\updownarrow c_2 & & d_3 \Big\updownarrow c_3 & & d_4 \Big\updownarrow c_4 & & \\[4pt]
u_1^\star & & u_2^\star & & u_3^\star & & u_4^\star & &
\end{array}
\tag{6}
$$

Inspection of this scheme reveals that it is similar in form to the DDT model analyzed in **Figure 2 and 3**: the reversible detachment step simply adds an additional transient state in each compartment. As we noted in the DDT model, cargo distributions can match demand over time with arbitrarily low error (see **Equation 4**). However, transport delays still exist. While releasing cargo to the wrong location is not an irreversible error, it slows delivery by temporarily arresting movement – known as a *diffusive trap* (see e.g. **Bressloff and Earnshaw, 2007**).

We found that cargo recycling creates a new tradeoff between convergence time and excess cargo left on the microtubules. Models that deliver a high percentage of their cargo ($c_i > d_i$) converge more slowly since they either release cargo into the diffusive traps (**Figure 7A1**) or have a slow detachment process (**Figure 7A2**). Models that deliver a low percentage of their cargo ($d_i > c_i$) converge quickly since they release little cargo into diffusive traps, allowing cargo to travel along the microtubules and reach all destinations within the neuron (**Figure 7A3**). **Figure 7B** shows the convergence of the three examples (A1, A2 and A3) over time. **Figure 7C** shows that the new tradeoff between cargo utilization and convergence time is similarly severe to the speed-accuracy tradeoff in

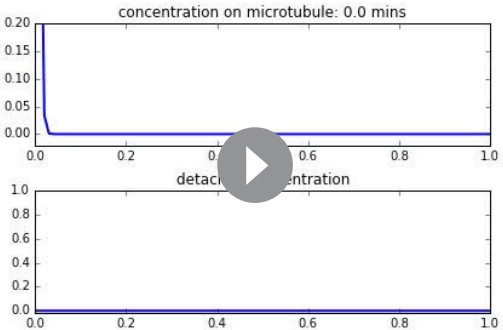
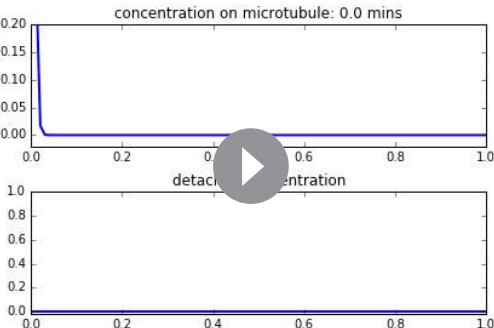
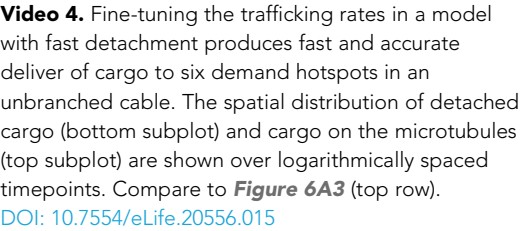

**Video 4.** Fine-tuning the trafficking rates in a model with fast detachment produces fast and accurate deliver of cargo to six demand hotspots in an unbranched cable. The spatial distribution of detached cargo (bottom subplot) and cargo on the microtubules (top subplot) are shown over logarithmically spaced timepoints. Compare to *Figure 6A3* (top row).

**Video 5.** The model fine-tuned for fast and accurate deliver of cargo to six demand hotspots misallocates cargo to three demand hotspots. The spatial distribution of detached cargo (bottom subplot) and cargo on the microtubules (top subplot) are shown over logarithmically spaced timepoints. Compare to *Figure 6A3* (middle row).

the sushi-belt model without reattachment. Models with reattachment that utilize cargo efficiently (for example, *Figure 7A2*) converge on similarly slow timescales to models without reattachment that deliver cargo accurately (for example, *Figure 6A1*). Models with less than 10% excess cargo required more than a day to reach steady-state within a tolerance of 10% mean error. On the other hand, models that converged around $10^3$ minutes (17 hr) required more than 90% of cargo to remain in transit at steady-state (*Figure 7C*).

## Distinct cell-type morphologies face order of magnitude differences in speed, precision and efficiency of trafficking

To establish the biological significance of these findings, we examined tradeoffs between speed, precision and excess cargo in reconstructed morphologies of five neuron cell types, spanning size and dendritic complexity (*Figure 8A*). We simulated trafficking and delivery of cargo to a spatially uniform target distribution in each cell type to reveal morphology-dependent differences. In all cases we used optimistic estimates of transport kinetics, corresponding to a diffusion coefficient of 10 $\mu m^2$ $s^{-1}$ (the rate constants were normalized to compartment size as in *Figure 6—figure supplement 1*).

*Figure 8B* shows spatial plots of the distribution of cargo on the microtubules ($u_i$, cyan-to-magenta colormap) and the distribution of delivered cargo ($u_i^{\star}$, black-to-orange colormap) for a model with an irreversible detachment rate of 8 $\times$ $10^{-5}$ $s^{-1}$. These parameters produce a relatively slow release of cargo: for each morphology, a sizable fraction of the cargo remains on the microtubules at ~3 hr, and it takes ~1–2 days to release all of the cargo. While the speed of delivery is roughly equivalent, the accuracy varied across the neural morphologies. The hippocampal granule cell converged to very low error (~11.7% mean error), while the larger L5 pyramidal cell converged to ~27.7% error. The smaller, but more elaborately branched, Purkinje cell converged to a similarly high average error of ~29.1%.

As before, faster detachment rates produce faster, but less accurate, delivery; while slower detachment rates produce more accurate, but slower, delivery. These tradeoffs across the entire family of regimes are plotted in *Figure 8C* (left). Adding a reattachment process largely preserved the effect of morphology on transport tradeoffs (*Figure 8C*, right). We fixed the detachment rate to be fast, since fast detachment produced the most favorable tradeoff in *Figure 7C*. Tradeoffs between excess cargo and speed of delivery emerged as the reattachment rate was varied (*Figure 8C*, right) and were more severe for the Purkinje cell and L5 pyramidal cell, and least severe for the Granule cell. Morphology itself therefore influences the relationship between delivery speed and precision, and/or excess cargo required, suggesting that different cell types might benefit from different trafficking strategies.

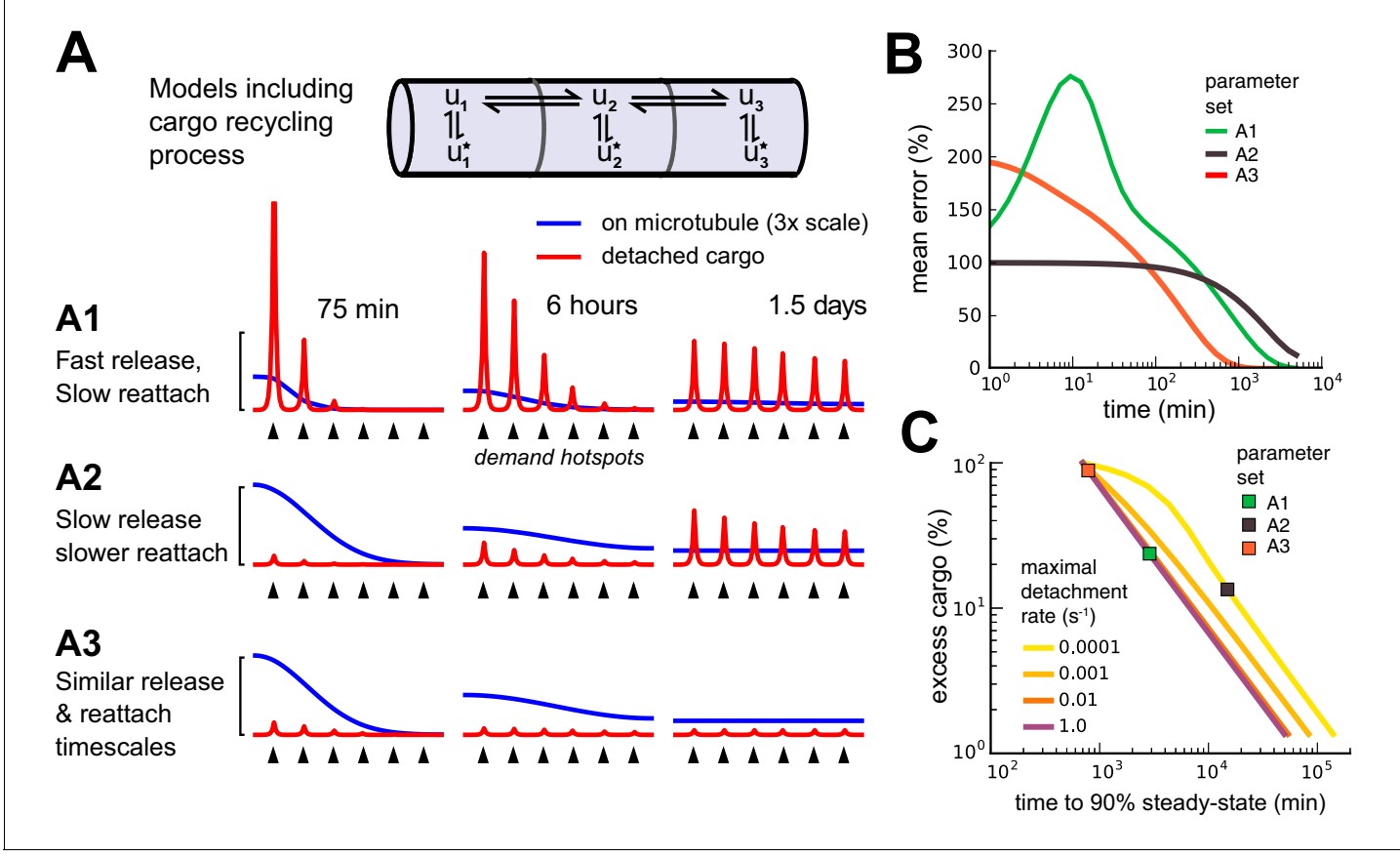

**Figure 7.** Adding a mechanism for cargo reattachment produces a further tradeoff between rate of delivery and excess cargo. (**A**) Simulations of three models (**A1, A2, A3**) with cargo recycling. As in *Figure 6*, cargo is distributed to six demand hotspots (black arrows). The distributions of cargo on the microtubules ($u_i$, blue) and detached cargo ($u_i^\star$, red) are shown at three times points for each model. (**B**) Mean percent error in the distribution of detached cargo as a function of time for the three models in panel A. (**C**) Tradeoff curves between excess cargo and time to convergence to steady-state (within 10% mean error across compartments) for fixed cargo detachment timescales (line color). For all detachment timescales, varying the reattachment timescale produced a tradeoff between excess cargo (fast reattachment) and slow convergence (slow reattachment). Colored squares denote the position of the three models in panel A.

## Discussion

The molecular motors that drive intracellular transport are remarkably efficient, achieving speeds of approximately 15 µm per minute (*Rogers and Gelfand, 1998*; *Dynes and Steward, 2007*; *Müller et al., 2008*). A naïve calculation based on this figure might suggest that subcellular cargo can be delivered precisely within a few hours in most dendritic trees. However, this ignores the stochastic nature of biochemical processes – motors spontaneously change directions and cargo can be randomly delivered to the wrong site. Such chance events are inevitable in molecular systems, and in the case of active transport they lead to diffusion of bulk cargo in addition to directed movement. If this kind of biochemical stochasticity played out in the sushi restaurant analogy, then the waiting time for a dish wouldn't simply equate to the time taken for the chef to prepare the dish and for the belt to convey it. Instead, the restaurant would be beleaguered by fickle customers who pick up dishes they do not want, either withholding them for an indefinite period, or setting them on another belt destined for the kitchen.

Mathematical models provide a rigorous framework to test the plausibility and the inherent relationships in conceptual models. Our study formalized the foremost conceptual model of dendritic transport (*Doyle and Kiebler, 2011*) to account for trafficking in realistic dendritic morphologies. Over a wide range of assumptions the model exhibits inherent and surprisingly punishing trade-offs between the accuracy of cargo delivery and the time taken to transport it over these morphologies.

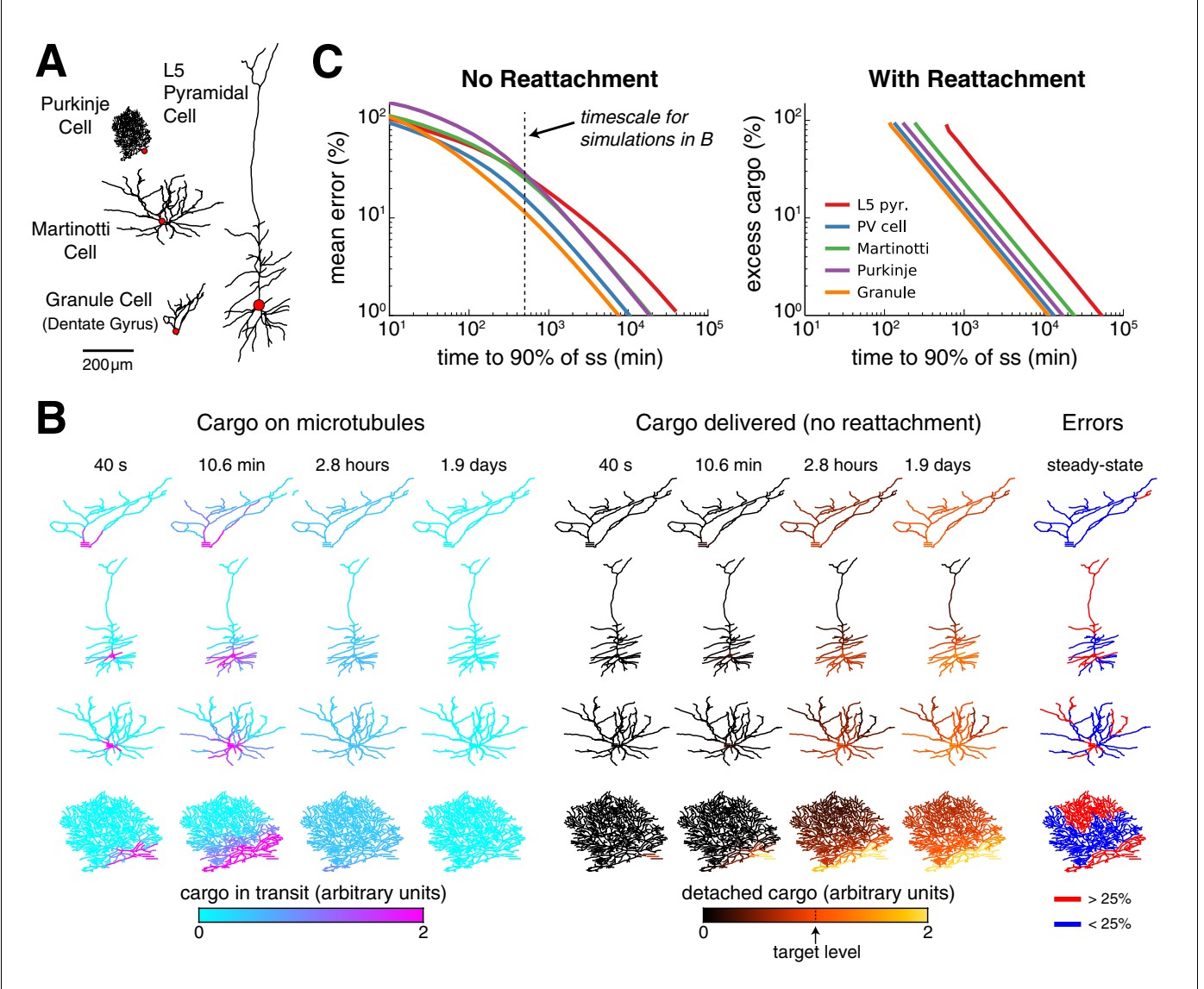

**Figure 8.** Effect of morphology on trafficking tradeoffs. (A) Representative morphologies from four neuron types, drawn to scale. The red dot denotes the position of the soma (not to scale). (B) Distribution of cargo on the microtubles ($u_i$) and delivered cargo ($u_i^\star$) at four time points for sushi-belt model with irreversible detachment. Cargo originated in the soma and was transported to a uniform distribution (all $a_i = b_i$, normalized to a diffusion coefficient of 10 $\mu m^2 \, s^{-1}$); the detachment rate was spatially uniform and equal to $8 \times 10^{-5} \, s^{-1}$. (C) Tradeoff curves for achieving a uniform distribution of cargo in realistic morphologies (PV cell = parvalbumin interneuron, morphology not shown). The sushi-belt model without reattachment (as introduced in *Figure 4*) suffers a tradeoff in speed and accuracy, while including reattachment (as in *Figure 7*) produces a similar tradeoff between speed and excess 'left-over' cargo. An optimistic diffusion coefficient of 10 $\mu m^2 \, s^{-1}$ was used in both cases. For simulations with reattachment, the detachment rate ($c_i$) was set equal to trafficking rates ($a_i, b_i$) for a one micron compartment. The dashed line denotes the convergence timescale for all simulations in panel B.

Using conservative estimates based on experimental data, the canonical sushi-belt model predicts delays of many hours or even days to match demand within 10%. Producing excess cargo and permitting reversible detachment from the microtubules can mitigate this tradeoff, but at a substantial metabolic cost, since a large amount of excess cargo is required.

These predictions are unsettling, because nucleus-to-synapse transport appears to play a role in time-critical processes. Elevated synaptic activity can initiate distal metabolic events including transcription (*Kandel, 2001*; *Deisseroth et al., 2003*; *Greer and Greenberg, 2008*; *Ch'ng et al., 2011*)

and this has been shown to be an important mechanism of neuronal plasticity (*Nguyen et al., 1994*; *Frey and Morris, 1997, 1998*; *Bading, 2000*; *Kandel, 2001*; *Redondo and Morris, 2011*). Moreover, neuronal activity has been observed to influence trafficking directly through second-messengers (*Mironov, 2007*; *Wang and Schwarz, 2009*; *Soundararajan and Bullock, 2014*), consistent with the hypothesis that trafficking rates are locally controlled. Genes that are transcribed in response to elevated activity can regulate synaptic strengths (*Flavell and Greenberg, 2008*; *Bloodgood et al., 2013*; *Spiegel et al., 2014*), and it has been suggested that nucleus-to-synapse trafficking of Arc directly regulates synaptic plasticity (*Okuno et al., 2012*). None of these findings imply that all kinds of molecular cargo are transported from the soma to distal dendritic locations, since mRNA can be sequestered and locally translated within dendrites (*Kang and Schuman, 1996*; *Cajigas et al., 2012*; *Holt and Schuman, 2013*). However, the speed, precision and efficiency trade-offs revealed in the sushi belt model provide a principled way to understand why some processes might require local biosynthesis, while others operate globally.

The different ways that local demand signals can influence trafficking and detachment can impact global performance, sometimes non-intuitively. Many of these effects should be experimentally testable. For example, transport bottlenecks can be induced if demand signals target local trafficking rates along microtubules (the DDT model). Transport to distal compartments will be substantially faster when proximal demand is introduced (see *Figure 3*). On the other hand, uniform trafficking combined with locally controlled detachment (DDD model, *Figure 4D*) can avoid bottlenecks, and often leads to faster transport. However, this is not always the case, as was shown in *Figure 5D*, where uniform trafficking is slower/inaccurate because cargo explores the basal dendritic tree even though there is no demand in that region. Spatial tuning of trafficking speed permitted more efficient cargo delivery in the model (see *Figure 6*). However, this has yet to be observed experimentally and would require extremely stereotyped morphology and physiological needs for it to be effective.

Intuitively, speed/precision tradeoffs arise because there is a conflict between exploring the dendritic tree and capturing cargo in specific locations. For irreversible cargo detachment, the capture rate needs to be roughly an order of magnitude slower than trafficking, otherwise, compartments proximal to the soma receive disproportionately high levels of cargo. This scaling is unfavorable for achieving high accuracy: if it takes roughly 100 min to distribute cargo throughout the dendrites, it will take roughly 1000 min (16–17 hr) before the cargo dissociates and is delivered to the synapses. If, instead, cargo is able to reattach, then fast reattachment favors exploration at the cost of greater excess (i.e. non-utilized) cargo, while slow reattachment hinders transport, since more cargo is detached and thus immobile. Even when the vast majority of cargo is produced as excess, global delivery times of several hours persist. Furthermore, if a neuron needs to rapidly replace a cargo that is already present in high concentrations, the strategy of generating excess cargo will result in large dilution times.

Overall, our results show that there are multiple ways that neurons can distribute cargo, but each differs in its speed, accuracy and metabolic cost. Therefore, optimizing for any one of these properties comes at the expense of the others. For example, in the model without reattachment (*Figure 4*), the same distribution of cargo can be achieved by: (a) location-dependent trafficking followed by uniform release, (b) uniform trafficking followed by location-dependent release, or (c) a mixture of these two strategies. Experimental findings appear to span these possibilities. (*Kim and Martin, 2015*) identified three mRNAs that were uniformly distributed in cultured *Aplysia* sensory neurons, but were targeted to synapses at the level of protein expression by localized translation (supporting option b). In contrast, the expression of *Arc* mRNA is closely matched to the pattern of Arc protein in granule cells of the dentate gyrus (possibly supporting option a; *Steward et al., 1998*; *Farris et al., 2014*; *Steward et al., 2014*). Trafficking kinetics do not just differ according to cargo identity – the same type of molecular cargo can exhibit diverse movement statistics in single-particle tracking experiments (*Dynes and Steward, 2007*). These differences lead us to speculate that different neuron types and different cargoes have adapted trafficking strategies that match performance tradeoffs to biological needs.

It is possible that active transport in biological neurons will be more efficient and flexible than models predict. Real neurons might use unanticipated mechanisms, such as a molecular addressing system, or nonlinear interactions between nearby cargo particles, to circumvent the tradeoffs we observed. For this reason, it is crucial to explore, quantitatively, the behavior of existing conceptual

models by replacing words with equations so that we can see where discrepancies with biology might arise. More generally, conceptual models of subcellular processes deserve more quantitative attention because they can reveal non-obvious constraints, relationships and connections to other biological and physical phenomena (*Smith and Simmons, 2001*; *Bressloff, 2006*; *Fedotov and Méndez, 2008*; *Newby and Bressloff, 2010b*; *Bhalla, 2011*; *Bressloff and Newby, 2013*; *Bhalla, 2014*). Other modelling studies have focused on the effects of stochasticity and local trapping of cargo on a microscopic scale, particularly in the context of low particle numbers (*Bressloff, 2006*; *Bressloff and Earnshaw, 2007*; *Fedotov and Méndez, 2008*; *Newby and Bressloff, 2010b*; *Bressloff and Newby, 2013*). We opted for a coarse-grained class of models in order to examine transport and delivery across an entire neuron. The model we used is necessarily an approximation: we assumed that cargo can be described as a concentration and that the multiple steps involved in cellular transport can lumped together in a mass action model.

By constraining trafficking parameters based on prior experimental measurements, we revealed that a leading conceptual model predicts physiologically important tradeoffs across a variety of assumptions. Experimental falsification would prompt revision of the underlying models as well as our conceptual understanding of intracellular transport. On the other hand, experimental confirmation of these tradeoffs would have fundamental consequences for theories of synaptic plasticity and other physiological processes that are thought to require efficient nucleus-to-synapse trafficking.

## Materials and methods

All simulation code is available online: https://github.com/ahwillia/Williams-etal-Synaptic-Transport

### Model of single-particle transport

Let $x_n$ denote the position of a particle along a 1-dimensional cable at timestep $n$. Let $v_n$ denote the velocity of the particle at timestep $n$; for simplicity, we assume the velocity can take on three discrete values, $v_n = \{-1, 0, 1\}$, corresponding to a retrograde movement, pause, or anterograde movement. As a result, $x_n$ is constrained to take on integer values. In the memoryless transport model (top plots in *Figure 1B, D and F*), we assume that $v_n$ is drawn with fixed probabilities on each step. The update rule for position is:

$$x_{n+1} = x_n + v_n$$

$$v_{n+1} = \begin{cases} -1 & \text{with probability } p_- \\ 0 & \text{with probability } p_0 \\ 1 & \text{with probability } p_+ \end{cases}$$

We chose $p_- = 0.2$, $p_0 = 0.35$ and $p_+ = 0.45$ for the illustration shown in *Figure 1*. For the model with history-dependence (bottom plots in *Figure 1B, D and F*), the movement probabilities at each step depend on the previous movement. For example, if the motor was moving in an anterograde direction on the previous timestep, then it is more likely to continue to moving in that direction in the next time step. In this case the update rule is written in terms of conditional probabilities:

$$v_{n+1} = \begin{cases} -1 & \text{with probability } p(-|v_n) \\ 0 & \text{with probability } p(0|v_n) \\ 1 & \text{with probability } p(+|v_n) \end{cases}$$

In the limiting (non-stochastic) case of history-dependence, the particle always steps in the same direction as the previous time step.

|  | $v_n = -1$ | $v_n = 0$ | $v_n = 1$ |
|---|---|---|---|
| $p(v_{n+1} = -1)$ | 1 | 0 | 0 |
| $p(v_{n+1} = 0)$ | 0 | 1 | 0 |
| $p(v_{n+1} = 1)$ | 0 | 0 | 1 |

We introduce a parameter $k \in [0, 1]$ to linearly interpolate between this extreme case and the memoryless model.

$$\begin{array}{c|ccc} & v_n = -1 & v_n = 0 & v_n = 1 \\ \hline p(v_{n+1} = -1) & p_-(1-k)+k & p_-(1-k) & p_-(1-k) \\ p(v_{n+1} = 0) & p_0(1-k) & p_0(1-k)+k & p_0(1-k) \\ p(v_{n+1} = 1) & p_+(1-k) & p_+(1-k) & p_+(1-k)+k \end{array} \tag{7}$$

The bottom plots of *Figure 1B and D* were simulated with $k = 0.5$.

To estimate the concentration and spatial distribution of cargo in real units, we used a 1 µm/s particle velocity and a 1 s time step to match experimental estimates of kinesin (*Klumpp and Lipowsky, 2005*, and references). We assumed a dendritic diameter of 7.2705 µm.

## Relationship of single-particle transport to the mass-action model

The mass-action model (*Equation 1*, in the Results) simulates the bulk movement of cargo across discrete compartments. Cargo transfer is modelled as an elementary chemical reaction obeying mass-action kinetics (*Keener and Sneyd, 1998*). For an unbranched cable, the change in cargo in compartment $i$ is given by:

$$\dot{u}_i = au_{i-1} + bu_{i+1} - (a+b)u_i \tag{8}$$

For now, we assume that the anterograde and retrograde trafficking rate constants ($a$ and $b$, respectively) are spatially uniform.

The mass-action model can be related to a drift-diffusion partial differential equation (*Figure 1E*) by discretizing into spatial compartments of size $\Delta$ and expanding around some position, $x$:

$$\dot{u}(x) \approx a\left[u(x) - \Delta\frac{\partial u}{\partial x} + \frac{\Delta^2}{2}\frac{\partial^2 u}{\partial x^2}\right] + b\left[u(x) + \Delta\frac{\partial u}{\partial x} + \frac{\Delta^2}{2}\frac{\partial^2 u}{\partial x^2}\right] - (a+b)u(x) \tag{9}$$

$$= a\left[-\Delta\frac{\partial u}{\partial x} + \frac{\Delta^2}{2}\frac{\partial^2 u}{\partial x^2}\right] + b\left[\Delta\frac{\partial u}{\partial x} + \frac{\Delta^2}{2}\frac{\partial^2 u}{\partial x^2}\right] \tag{10}$$

We keep terms to second order in $\Delta$, as these are of order $dt$ in the limit $\Delta \to 0$ (*Gardiner, 2009*). This leads to a drift-diffusion equation:

$$\dot{u}(x) = \frac{\partial u}{\partial t} = \underbrace{(b-a)}_{\text{drift coefficient}}\frac{\partial u}{\partial x} + \underbrace{\left(\frac{a+b}{2}\right)}_{\text{diffusion coefficient}}\frac{\partial^2 u}{\partial x^2} \tag{11}$$

Measurements of the mean and mean-squared positions of particles in tracking experiments, or estimates of the average drift rate and dispersion rate of a pulse of labeled particles can thus provide estimates of parameters $a$ and $b$.

How does this equation relate to the model of single-particle transport (*Figure 1A–B*)? For a memoryless biased random walk, the expected position of a particle after $n$ time steps is $E[x_n] = n(p_+ - p_-)$ and the variance in position after $n$ steps is $n\left(p_+ + p_- - (p_+ - p_-)^2\right)$. For large numbers of non-interacting particles the mean and variance calculations for a single particle can be directly related to the ensemble statistics outlined above. We find:

$$a = \frac{2p_+ - (p_+ - p_-)^2}{2}$$

$$b = \frac{2p_- - (p_+ - p_-)^2}{2}$$

This analysis changes slightly when the single-particle trajectories contain long, unidirectional runs. The expected position for any particle is the same $E[x_n] = n(p_+ - p_-)$; the variance, in contrast, increases as run lengths increase. However, the mass-action model can often provide a good fit in this regime with appropriately re-fit parameters (see *Figure 1F*). Introducing run lengths produces a larger effective diffusion coefficient and thus provides faster transport. As long as the single-particles have stochastic and identically distributed behavior, the ensemble will be well-described by a normal

distribution by the central limit theorem. This only breaks down in the limit of very long unidirectional runs, as the system is no longer stochastic (*Figure 1—figure supplement 1*).

## Stochastic interpretation of the mass-action model

An important assumption of the mass-action model is that there are large numbers of transported particles, so that the behavior of the total system is deterministic. Intuitively, when each compartment contains many particles, then small fluctuations in particle number don't appreciably change concentration. Many types of dendritic cargo are present in high numbers (*Cajigas et al., 2012*).

When few cargo particles are present, fluctuations in particle number are more functionally significant. Although we did not model this regime directly, the mass-action model also provides insight into this stochastic regime. Instead of interpreting $u_i$ as the amount of cargo in compartment $i$, this variable (when appropriately normalized) can be interpreted as the probability of a particle occupying compartment $i$. Thus, for a small number of transported cargoes, the mass-action model describes the average, or expected, distribution of the ensemble.

In this interpretation, the mass-action model models a spatial probability distribution. Let $p_i$ denote the probability of a particle occupying compartment $i$. If a single particle starts in the somatic compartment at $t = 0$, and we query this particle's position after a long period of transport, then the probability ratio between of finding this particle in any parent-child pair of compartments converges to:

$$\left.\frac{p_p}{p_c}\right|_{ss} = \frac{b}{a}$$

which is analogous to *Equation (3)* in the Results.

In the stochastic model, the number of molecules in each compartment converges to a binomial distribution at steady-state; the coefficient of variation in each compartment is given by:

$$\sqrt{\frac{1 - p_i^{(ss)}}{n p_i^{(ss)}}}$$

This suggests two ways of decreasing noise. First, increasing the total number of transported molecules, $n$, proportionally decreases the noise by a factor of $1/\sqrt{n}$. Second, increasing $p_i$ decreases the noise in compartment $i$. However, this second option necessarily comes at the cost of decreasing occupation probability and thus increasing noise in other compartments.

## Estimating parameters of the mass-action model using experimental data

The parameters of the mass-action model we study can be experimentally fit by estimating the drift and diffusion coefficients of particles over the length of a neurite. A common approach is to plot the mean displacement and mean squared displacement of particles as a function of time. The slopes of the best-fit lines in these cases respectively estimate the drift and diffusion coefficients. Diffusion might not accurately model particle movements over short time scales because unidirectional cargo runs result in superdiffusive motion, evidenced by superlinear increases in mean squared-displacement with time (*Caspi et al., 2000*). However, over longer timescales, cargoes that stochastically change direction can be modelled as a diffusive process (*Soundararajan and Bullock, 2014*).

The mass-action model might also be fitted by tracking the positions of a population of particles with photoactivatable GFP (*Roy et al., 2012*). In this case, the distribution of fluorescence at each point in time could be fit by a Gaussian distribution; the drift and diffusion coefficients are respectively proportional to the rate at which the estimated mean and variance evolves over time.

These experimental measurements can vary substantially across neuron types, experimental conditions, and cargo identities. Therefore, in order to understand fundamental features and constraints of the sushi belt model across systems, it is more useful to explore relationships within the model across ranges of parameters. Unless otherwise stated, the trafficking kinetics were constrained so that $a_i + b_i = 1$ for each pair of connected compartments. This is equivalent to having a constant diffusion coefficient of one across all compartments. Given a target expression pattern along the microtubules, this is the only free parameter of the trafficking simulations; increasing the diffusion

coefficient will always shorten convergence times, but not qualitatively change our results. In *Figures 6–8* we fixed the diffusion coefficient to an optimistic value of 10 μm² s⁻¹ based on experimental measurements (*Caspi et al., 2000*; *Soundararajan and Bullock, 2014*) and the observation that long run lengths can increase the effective diffusion coefficient (*Figure 1—figure supplement 1*).

## Steady-state analysis

The steady-state ratio of trafficked cargo in neighboring compartments equals the ratio of the trafficking rate constants (*Equation 2*). Consider an unbranched neurite with non-uniform anterograde and retrograde rate constants (*Equation 1*). It is easy to verify the steady-state relationship in the first two compartments, by setting $\dot{u}_1 = 0$ and solving:

$$-a_1 u_1 + b_1 u_2 = 0 \Rightarrow \left.\frac{u_1}{u_2}\right|_{ss} = \frac{b_1}{a_1}$$

Successively applying the same logic down the cable confirms the condition in *Equation 2* holds globally. The more general condition for branched morphologies can be proven by a similar procedure (starting at the tips and moving in).

It is helpful to re-express the mass-action trafficking model as a matrix differential equation, $\dot{\mathbf{u}} = A\mathbf{u}$, where $\mathbf{u} = [u_1, u_2, ...u_N]^T$ is the state vector, and $A$ is the state-transition matrix. For a general branched morphology, $A$ will be nearly tridiagonal, with off-diagonal elements corresponding to branch points; matrices in this form are called Hines matrices (*Hines, 1984*). For the simpler case of an unbranched cable, $A$ is tridiagonal:

$$A = \begin{bmatrix} -a_1 & b_1 & 0 & & \cdots & & 0 \\ a_1 & -b_1-a_2 & b_2 & 0 & & & \\ 0 & a_2 & -b_2-a_3 & b_3 & \ddots & & \vdots \\ \vdots & 0 & a_3 & \ddots & & & 0 \\ & & \ddots & & & -b_{N-2}-a_{N-1} & b_{N-1} \\ 0 & & \cdots & & 0 & a_{N-1} & -b_{N-1} \end{bmatrix}$$

For both branched and unbranched morphologies, each column of $A$ sums to zero, which reflects conservation of mass within the system. Assuming nonzero trafficking rates, the rank of $A$ is exactly $N-1$ (this can be seen by taking the sum of the first $N-1$ rows, which results in $-1$ times the final row). Thus, the nullspace of $A$ is one-dimensional. *Equation (3)* describes this manifold of solutions: the level of cargo can be scaled by a common multiplier across all compartments without disrupting the relation in (2).

The steady-state distribution, $\tilde{\mathbf{u}}$, is a vector that spans the nullspace of $A$. It is simple to show that all other eigenvalues $A$ are negative using the Gershgorin circle theorem; thus, the fixed point described by *Equation 2* is stable. The convergence rate is determined by the non-zero eigenvalue with the smallest magnitude of $A$. There are no other fixed points or limit cycles in this system due to the linearity of the model.

## Biologically plausible model of a local demand signal

There are many biochemical mechanisms that could signal demand. Here we briefly explore cytosolic calcium, $[Ca]_i$, as a candidate mechanism since it is modulated by local synaptic activity and $[Ca]_i$ transients simultaneously arrest anterograde and retrograde microtubular transport for certain cargoes (*Wang and Schwarz, 2009*). We represent the effect of the calcium-dependent pathway by some function of calcium, $f([Ca_i])$. This function could, for example, capture the binding affinity of $[Ca]_i$ to enzymes that alter the kinetics of motor proteins; the Hill equation would provide a simple functional form. If all outgoing trafficking rates of a compartment are controlled by cytosolic calcium — i.e. for any parent-child pair of compartments we have $a = f([Ca]_p)$ and $b = f([Ca]_c)$ — then condition in *Equation 4* is satisfied:

$$\frac{b}{a} = \frac{f([Ca]_c)}{f([Ca]_p)} = \frac{\tilde{u}_p}{\tilde{u}_c} \tag{12}$$

where $\tilde{u}_i = 1/f([Ca]_i)$. We emphasize that other potential signalling pathways could achieve the same effect, so while there is direct evidence for $[Ca]_i$ as an important signal, the model can be interpreted broadly, with $[Ca]_i$ serving as a placeholder for any local signal identified experimentally. Further, $[Ca]_i$ itself may only serve as a demand signal over short timescales, while other, more permanent, signals such as microtubule-associated proteins (*Soundararajan and Bullock, 2014*) are needed to signal demand over longer timescales.

## Simulations in realistic morphologies

We used a custom-written Python library to generate movies and figures for all simulations in realistic morphologies (*Williams, 2016*). We obtained the CA1 pyramidal cell model from the online repository ModelDB (*Hines et al., 2004*), accession number 144541 (*Migliore and Migliore, 2012*). We used the default spatial compartments and set the trafficking and dissociation parameters of the mass-action transport model without reference to the geometry of the compartments. Model simulations were exact solutions using the matrix exponential function from the SciPy library at logarithmically spaced timepoints (*Jones et al., 2001*). In *Figure 2* we simulated electrical activity of this model with excitatory synaptic input for 5 s using the Python API to NEURON (*Hines et al., 2009*). We used the average membrane potential over this period to set the target demand level. In *Figures 3* and *4*, we imposed artificial demand profiles with regions of low-demand and high-demand (an order-of-magnitude difference) as depicted in the figures. Time units for simulations of the CA1 model were were normalized by setting trafficking rates $a_i + b_i = 1$ (which corresponds to a unit diffusion coefficient).

In *Figure 8*, we obtained representative morphologies of five cell types from neuromorpho.org (*Ascoli et al., 2007*). Specifically, we downloaded a Purkinje cell (Purkinje-slice-ageP43-6), a parvalbumin-positive interneuron (AWa80213), a Martinotti cell (C100501A3), a layer-5 pyramidal cell (32-L5pyr-28), and a granule cell from the dentate gyrus (041015-vehicle1). In these simulations, we scaled the trafficking parameters inversely proportional to the squared distance between the midpoints of neighboring compartments, which is mathematically appropriate to keep the (approximated) diffusion coefficient constant across the neural morphology. We confirmed that compartment size had minimal effects on the convergence rate and steady-state cargo distribution when the trafficking rates were scaled in this way in the reduced cable model (*Figure 6—figure supplement 1*).

For simulations with reattachment in *Figure 8*, we set the detachment rate ($c_i$) equal to the trafficking rates ($a_i, b_i$) for a one micron compartment. We did this based on the observation that a fast detachment rate provided the most favorable tradeoff curve in *Figure 7C*.

## Incorporating detachment and reattachment into the mass-action model

For compartment $i$ in a cable, the differential equations with detachment become:

$$\dot{u}_i = a_{i-1}u_{i-1} - (a_i + b_{i-1} + c_i)u_i + b_iu_{i+1}$$
$$\dot{u}_i^\star = c_iu_i$$

When $a_i, b_i \gg c_i$, then the distribution of cargo on the microtubules ($u_i$) approaches a quasi-steady-state that follows *Equation 3*. In *Figure 4*, we present DDT and DDD models as two strategies that distribute cargo to match a demand signal $\tilde{u}_i^\star$. As mentioned in the main text, a spectrum of models that interpolate between these extremes are possible. To interpolate between these strategies, let $F$ be a scalar between 0 and 1, and let $\tilde{u}^\star$ be normalized to sum to one. We choose $a_i$ and $b_i$ to achieve:

$$\tilde{u}_i = F\tilde{u}_i^\star + (1-F)/N$$

along the microtubular network and choose $c_i$ to satisfy

$$c_i \propto \frac{\tilde{u}_i^\star}{F\tilde{u}_i^\star + (1-F)/N}$$

Here, $N$ is the number of compartments in the model. Setting $F = 1$ results in the DDT model (demand is satisfied purely by demand-modulated trafficking, and non-specific detachment,

*Figure 4C*). Setting $F = 0$ results in the DDD model (demand is satisfied purely by demand-modulated detachment, and uniform/non-specific trafficking, *Figure 4D*). An interpolated strategy is shown in *Figure 4E* ($F = 0.3$).

The mass-action model with reattachment (*Equation 6*) produces the following system of differential equations for a linear cable, with $d_i$ denoting the rate constant of reattachment in compartment $i$

$$\begin{aligned} \dot{u}_i &= a_{i-1}u_{i-1} - (a_i + b_{i-1} + c_i)u_i + b_i u_{i+1} + d_i u_i^\star \\ \dot{u}_i^\star &= c_i u_i - d_i u_i^\star \end{aligned}$$

We examined the DDD model with $N = 100$ compartments and diffusion coefficient of 10 μm²s⁻¹. The maximal detachment rate constant and the reattachment rates were tunable parameters, while the reattachment rates were spatially uniform. Results were similar when reattachment was modulated according to demand (data not shown, see supplemental simulations at https://github.com/ahwillia/Williams-etal-Synaptic-Transport).

## Globally tuning transport rates to circumvent the speed-specificity tradeoff

In *Figure 6*, we explored whether fine-tuning the trafficking rates could provide both fast and precise cargo distribution. We investigated the DDD model with fast detachment rates in an unbranched cable with equally spaced synapses and $N = 100$ compartments. Large detachment rates produced a proximal bias in cargo delivery which we empirically found could be corrected by setting the anterograde and retrograde trafficking rates to be:

$$a_i = \frac{D}{2} + \beta \cdot \frac{N-1-i}{N-2}$$

$$b_i = \frac{D}{2} - \beta \cdot \frac{N-1-i}{N-2}$$

where $i = \{1, 2, ...N-1\}$ indexes the trafficking rates from the soma ($i = 1$) to the other end of the cable ($i = N-1$), and $D = 10\,\mu m^2/s$ is the diffusion coefficient. Faster detachment rates require larger values for the parameter $\beta$; note that $\beta < D/2$ is a constraint to prevent $b_i$ from becoming negative. This heuristic qualitatively improved, but did not precisely correct for, fast detachment rates in the DDT model (data not shown).

Intuitively, the profile of the proximal delivery bias is roughly exponential (*Figure 6B*), and therefore the anterograde rates need to be tuned more aggressively near the soma (where the bias is most pronounced), and more gently tuned as the distance to the soma increases. Importantly, tuning the trafficking rates in this manner does not alter the diffusion coefficient along the length of the cable (since $a_i + b_i$ is constant by construction). These manipulations produce a nonzero drift coefficient to the model, which corrects for the proximal bias in cargo delivery.

## Acknowledgements

We thank Aoife McMahon, Lasani Wijetunge, Eve Marder, Subhaneil Lahiri, Friedemann Zenke, and Benjamin Regner for useful feedback on the manuscript, and thank Jeff Gelles and Simon Bullock for useful discussion. This research was supported by the Department of Energy Computational Science Graduate Fellowship, NIH Grant 1P01NS079419, NIH Grant P41GM103712, the Howard Hughes Medical Institute.

## Additional information

### Funding

| Funder | Grant reference number | Author |
| --- | --- | --- |
| Department of Energy Computational Science Graduate Fellowship | | Alex H Williams |

| Howard Hughes Medical Institute | | Terrence J Sejnowski |
| --- | --- | --- |
| National Institutes of Health | P41GM103712 | Terrence J Sejnowski |
| National Institutes of Health | 1P01NS079419 | Timothy O'Leary |

The funders had no role in study design, data collection and interpretation, or the decision to submit the work for publication.

## Author contributions

AHW, Conception and design, Acquisition of data, Analysis and interpretation of data, Drafting or revising the article; CO, TO, Conception and design, Analysis and interpretation of data, Drafting or revising the article; TJS, Analysis and interpretation of data, Drafting or revising the article

## Author ORCIDs

Cian O'Donnell, http://orcid.org/0000-0003-2031-9177

Terrence J Sejnowski, http://orcid.org/0000-0002-0622-7391

Timothy O'Leary, http://orcid.org/0000-0002-1029-0158

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
