## [Decision Letter]

[Editors’ note: a previous version of this study was rejected after peer review, but the authors submitted for reconsideration. The first decision letter after peer review is shown below.]

Thank you for submitting your work entitled "Dendritic trafficking faces physiologically critical speed-precision tradeoffs" for consideration by *eLife*. Your article has been favorably evaluated by Naama Barkai (Senior Editor) and three reviewers, one of whom is a member of our Board of Reviewing Editors. Our decision has been reached after consultation between the reviewers. Based on these discussions and the individual reviews below, we regret to inform you that your work will not be considered further for publication in *eLife*.

Here is a synthesis of the views of the reviewers, highlighting the main points.

1) All three reviewers appreciated the development of a general and simplecoarse-grained model of cargo localization, and the incorporation of severalaspects of experimentally observed trafficking.

2) There is a shared common concern about whether real neurons do experience the bottlenecks and very slow settling of cargo distributions that the model predicts. All three reviewers felt that the real settling time was likely tobe faster than the model predicts.

3) There were also shared concerns about the model assumptions, particularlywith respect to the mechanisms assumed for the system. The reviewers felt thatsome likely, and experimentally supported mechanisms were left out of theanalysis. Some of the mechanisms that the reviewers mentioned included cargo reloading, feedback, and having sufficiently large cargo traffic to feed transient local requirements for cargo. The reviewers felt that these mechanisms might eliminate the slow time-courses predicted by the current study.

4) The reviewers were hoping to see a more complete mapping to experimentswith stronger mechanistic predictions and closer ties to biologicalobservations.

*Reviewer #1:*

This study puts quantitative flesh on the bones of the sushi-belt model for transport in the dendrites and its interaction with local signals resulting in cargo offloading.

At the outset it is important to make the point that the sushi-belt model as originally proposed was a word-model, and the process of converting it to mathematical form is non-trivial and itself involves many mechanistic assumptions and insights. While I generally appreciate the motivation of the study, I have concerns about some of the assumptions and feel that some aspects of the model behavior and study conclusions may be artifacts of these assumptions. Specifically, I feel that the key conclusions about speed-specificity tradeoffs and the time-course for attaining desired distributions are overstated. Below I indicate how other model assumptions, or even factoring another term into the same model (point 6), may overcome the stated limitations in achieving target cargo distributions.

1) The first aspect of the model involves the microtubule-based transport.

Here the authors adapt and simplify a biased 1-D random-walk model by

Muller et al. They consider two variants: simple independent probabilities,and history-dependent probabilities. This first set of model conversion iswell-grounded in the literature and results in a familiar model form. Thistakes them to an analysis of cargo distribution as a function of transportrates, and they show that a variety of distributions can be achieved throughthis mechanism. If I understand correctly, the main novelty in this figureis the result that even with long runs the mass-action model fits thestochastic one reasonably well. It would be nice to have a panel of thisdependency, that is, a graph of fit vs. run length.

2) The next step of the modeling is to consider how the local rates oftransport might be modulated. The authors come up with a somewhat limitingmodel here, assuming only a single signal (Ca for convenience), and achieveforward/backward rate control by taking the signal levels in successive spatialcompartments. Here one could readily imagine that different localsignals might be a more versatile (and spatially more precise) way to achievecontrol of forward and backward rates of transport. Can the authors examinethis?

3) The electrical calculations in Figure 2 are poorly described. I assumethat the authors use the full Migliore 2012 model to obtain an electricalpotential distribution upon synaptic stimulation. It would have been usefulto have seen the electrical potential and its time-course. Over what timewas the 'average potential' taken? How did the potential map to the rates?

The authors refer to the methods section but there is insufficient informationthere.

4) The authors go through a few more elaborations in the model, before bringing in a 'detachment' scheme that finally takes their model to something more like the full sushi-belt model. In Figure 1–Figure 3 I am concerned that the analysis talks about density of cargo on the motors rather than free cargo in thedendrites. First, it would be valuable to make this distinction clearer tothe reader. Second, it would be valuable to discuss whether these predictionshave physiological observations to compare with. I do not have a sense forhow much cargo sits on the motors, and how much variability is observedin the distribution of motor-attached vs. detached forms.

5) When detachment is incorporated into the model, the authors find that onegets non-specific cargo delivery, as well as depletion of cargo attached tothe motors. I am concerned that these phenomena are more a reflection ofassumptions than physiology. Specifically, the unloading of cargo is anopen-loop, stimulus-driven process in the model. I wonder how many of thesefindings would hold if the unloading rate were driven not just by stimulus,but also by feedback based on amount of desired cargo that was alreadypresent. That is, a term dependent on ui*. Further, the degradation itselfcould also be driven by feedback. I suspect that the set point might bereached much faster with these elaborations.

6) Looking in more detail at the equations in the subsection “Incorporating detachment and degradation into the mass-action model”, I was trying to understand the effect of loading density of the cargo. Specifically,if ui is large and the desired ui* is small, surely the system shouldgive a very rapid convergence to the target ui*? In other words, if thereis a huge amount of cargo available and going past, then one can quicklyobtain what one needs in any location to a high degree of accuracy. Itseems to me that the loading term should also play a role in the analysis on

Figure 5 and Figure 6. Thus the 'slow detachment' case could actually be fastin absolute time terms if one were to factor in lots of available cargo.

I do not see this factor in the analysis in the subsection “Conservative experimental estimates of trafficking parameters suggest that the tradeoff between speed and specificity is severe”.

*Reviewer #2:*

In this manuscript, the authors developed a theoretical model for transport of cargoes on microtubules. Analytical solutions of the model show that such transport can either be fast or precise but not both. (Precision in this case means similarity to target cargo concentration at the destination.) In particular, the authors considered two different transport schemes were considered: (1) specific transport, uniform detachment, and (2) uniform transport, specific detachment.

1) A consequence of the first transport scheme is that bottlenecks will occur with the same probability in the main dendrites and the subsequent branch dendrites, since the rate constants within the whole neuron are modeled by the same function. However, it is reasonable to ask if neurons in reality do exhibit bottlenecks in the main dendrite and the branch/daughter dendrites at the same frequency.

2) Perhaps there could be more detailed studies of transports using a combination of these two transport schemes (Figure 4). For example, will an intermediate strategy improve speed and precision, i.e., can a scheme involving intermediate transport-specificity and intermediate detachment-specificity circumvent the problems of bottlenecks and cargo leakage? Perhaps a phase space plot that illustrates the effect of the combinations of schemes on accuracy and transport time may help to conveythe information better.

*Reviewer #3:*

The manuscript by Williams applies a "sushi-belt delivery model" to cargo transport In CA1 pyramidal neurons. The goal is to understand the tradeoff between speed versus precision during cargo transport along microtubules by motor proteins. The manuscript opens with a rather general discussion (mass-action model) of convection-diffusion in a channel that is coarse-grained at the level of adjacent boxes. This idea is extended to model pyramidal neurons where the steady state distribution of cargo is calculated w.r.t. a target profile. The authors then model a bottleneck situation by assuming low cargo transition rates (epsilon) into a compartment, and test how the system converges to steady state as a function of epsilon. The model is taken further by introducing detachment from microtubules (possibly followed by diffusion-recapture – details unclear), and looking at efficiency of transport under two possibilities – cargo is selectively transported to target or is uniformly distributed (combined with detachment).

The goal is laudable because it attempts to present a generalized and simple mathematically solvable coarse-grained description of cargo localization in a complex neuronal geometry. Most of the assumptions of the mathematical model appear valid, their rate constants seem to match the experimental velocities and they seem to have taken into consideration various scenarios during cellular transport. However, we feel that the paper starts off being rather general, and remains more-or-less so till the end. For example, Figure 4 show that the target cargo distributions are always achieved irrespective of the transport/detachment ratio. What is one expected to learn from this, and how might it be useful to plan future experiments? If the message is that many strategies can be employed to achieve target distributions, then this is a rather weak message unless this theme is developed further with specific examples and suggestions. Similarly, the observation (subsection “Convergence rate”, last paragraph) that transport will achieve steady state faster if bottlenecks are removed – why is this surprising? This part is followed up by a few poorly explained lines where the results (Figure 3) seem interesting, but are obscured by unnecessary usage of complicated Latin words. On the same lines, in the places where it is mentioned, the connection to experiments is rather weak. Whether these assumptions hold true in a biological setting has not been tested for any neuronal cargoes. Live imaging to show that at least a few cargoes follow this model would have helped.

The authors talk about detachment and degradation of cargoes. But how does reloading of cargoes occur in instances when continuous supply is required?

The authors suggest using their model that accurate transport is slower, and faster rates of transport requires much greater complexity and is very sensitive to perturbations. This is hard to visualize in case of most neuronal functions where efficient robust and rapid signaling does occur during processes such as long term memory formation, signaling at the synapse etc. Again, there appears to be a disconnection between real biology and the model.

Taken together, we feel that this work would not have sufficient impact to warrant publication in *eLife*. This is in contrast to models of microtubule transport (e.g. Lipowsky group PNAS paper) which have made more specific mechanistic predictions that advanced the field and inspired new experiments. We suggest the authors also improve the writing of this manuscript in consultation with some experimental colleagues. Perhaps addition of some experiments as preliminary tests of models would also help.

[Editors’ note: what now follows is the decision letter after the authors submitted for further consideration.]

Thank you for submitting your article "Dendritic trafficking faces physiologically critical speed-precision tradeoffs" for consideration by *eLife*. Your article has been favorably evaluated by Naama Barkai (Senior Editor) and three reviewers, one of whom is a member of our Board of Reviewing Editors. The following individual involved in review of your submission has agreed to reveal their identity: Ambarish Kunwar (Reviewer #3).

The reviewers have discussed the reviews with one another and the Reviewing Editor has drafted this decision to help you prepare a revised submission.

The reviewers all felt that the paper was much improved from the earlier version. They made some small suggestions for minor improvements and clarifications, not requiring further review.

Summary:

In this study Williams et al. explore simulations of the 'sushi-belt' family of models, including local signals to unload cargo. They use deterministic methods after comparing with some preliminary stochastic calculations. They embed the model into a detailed neuronal morphology. The authors exercise the model on some not-so-obvious predictions such as speed and accuracy tradeoffs.

Essential revisions:

1) I'm surprised that the feedback and cargo recycling processes do not bring rapid settling to the system without large cargo excess. I think this is one of the key findings of the paper. I would suggest that the authors move some of the panels from Figure 5—figure supplement 2 into the main body of the paper, so as to better present this result.

2) The authors have based their entire simulation on a real life neuron (CA1 Pyramidal Cell) with a fixed number of compartments (742). It would be good if the authors throw some light on (i) how sensitive their simulations are to the number of compartments in the neuron, and (ii) how the compartment size is related to the cargo size.

---

## [Author Response]

[Editors’ note: the author responses to the first round of peer review follow.]

*Here is a synthesis of the views of the reviewers, highlighting the main points.*

*1) All three reviewers appreciated the development of a general and simplecoarse-grained model of cargo localization, and the incorporation of severalaspects of experimentally observed trafficking.*

We are pleased that the reviewers recognized the need for a formalization of conceptual models in this field and an assessment of the relationships they imply. We further believe that studies like ours are important because researchers in related fields do not always appreciate the relevance of low-level biophysical processes to higher level physiological function, and because word models that often inform experimental designs and interpretations of data imply relationships that are not evident until the model is formalized. Not all of the reviewers appreciated these points.

*2) There is a shared common concern about whether real neurons do experience the bottlenecks and very slow settling of cargo distributions that the model predicts. All three reviewers felt that the real settling time was likely tobe faster than the model predicts.*

Our goal is to challenge intuitions about how existing trafficking models behave with actual numbers. Settling times have never been measured, as such we believe our findings, coupled with the useful suggestions from the reviewers, will advance the field by motivating attempts to actually measure settling times.

A) Either settling times are fast (and accurate) in biology, which calls into question the current working model of the field.

B) Or, nucleus-to-synapse trafficking is not as fast (or accurate) as expected by the reviewers and other experts in the field.

Either of these possibilities is significant. Determining which is correct will require a series of experimental studies, possibly in multiple different preparations, cell types, and species. Our model can be used to generate specific hypotheses for these experiments, which would be difficult to formulate otherwise. To aid this, we now discuss connection between the model and experiments more explicitly.

On the other hand, our model may have had flawed assumptions. The crucial concern raised in the reviews (that our model did not allow recirculation of cargo) has now been addressed. The other very broad suggestion was some kind of feedback. The model formulation assumes an accurate (perfect, in fact) mechanism that signals demand for cargo according to the actual quantity needed, but does so with fixed rates. We have clarified this in the revision.

Recycling of cargo provides a means to allow more rapid delivery of cargo to potential sites of demand. This does indeed speed up settling times, but with the additional metabolic cost of requiring excess cargo (see next point), and the best case still predicts several hours for typical morphologies.

Beyond such mechanisms, there is no evidence, to our knowledge, of any more sophisticated feedback mechanism, e.g. some kind of mechanism that routes cargo to specific dendrites. We discuss how feedback could optimize delivery times further, but we don't believe that it is profitable to iteratively tweak the model by postulating additional mechanisms that have yet to be observed. Instead, we believe it is far more useful to motivate experimental falsification of the predictions of a parsimonious model.

*3) There were also shared concerns about the model assumptions, particularlywith respect to the mechanisms assumed for the system. The reviewers felt thatsome likely, and experimentally supported mechanisms were left out of theanalysis. Some of the mechanisms that the reviewers mentioned included cargo reloading, feedback, and having sufficiently large cargo traffic to feed transient local requirements for cargo. The reviewers felt that these mechanisms might eliminate the slow time-courses predicted by the current study.*

As discussed in the previous point, we have now allowed for cargo reloading and discussed feedback. Cargo reloading introduces a new tradeoff between trafficking accuracy, speed and excess cargo. Two new figures are devoted to this (Figure 5—figure supplement 2 and new Figure 6). The excess cargo required to speed up settling times is surprisingly severe and still results in settling times of several hours.

A further point that we have now discussed is that excess cargo will lead to long dilution times if an existing species of cargo needs to be replaced. We did not model dilution explicitly as this would require a lot of guesswork and in any case could only serve to exacerbate the long settling times predicted by the existing model.

*4) The reviewers were hoping to see a more complete mapping to experimentswith stronger mechanistic predictions and closer ties to biologicalobservations.*

Our original manuscript contained several experimental predictions, but we recognize that these connections could have been made more clearly. We have substantially tightened the manuscript and made an effort to ensure that our major points and predictions are highlighted more prominently and clearly. We highlight three salient connections to experimental data below:

A) In Figure 1, along with details documented in the Methods and online code repository (https://github.com/ahwillia/Williams-etal-Synaptic-Transport), provide a mapping of single-particle movements to the parameters of the model we study.

B) Figure 3, in particular panel F, presents a series of direct experimental predictions for the DDT model related to trafficking bottlenecks. Candidate systems for testing these predictions include activity-dependent trafficking of *Arc* mRNA.

C) Figure 6 is completely new and provides a comprehensive summary of various model assumptions and predictions in realistic neuron morphologies.

Reviewer #3 commented that our results are “general” and that this is somehow a negative attribute. We are slightly puzzled by this assessment. Just as there are many kinds of experimental papers with different goals, there are many kinds of theory and modelling papers. Some, such as the PNAS paper the reviewer cites, focus on detailed molecular interactions. Others, such as ours, attempt to synthesize a complex, messy set of observations and biological questions into something intelligible that reveals high- level relationships. It is a mistake to think that a high-level focus doesn't apply because the object of study is biochemical in nature. We believe that both kinds of studies are important and that it is inappropriate to judge one type of study according to the attributes of a completely different study. Moreover, our study does, in fact, make numerous predictions that are testable in spite of their generality, as we discuss in detail below and in the revised manuscript.

*Reviewer #1:*

*This study puts quantitative flesh on the bones of the sushi-belt model for transport in the dendrites and its interaction with local signals resulting in cargo offloading.*

*At the outset it is important to make the point that the sushi-belt model as originally proposed was a word-model, and the process of converting it to mathematical form is non-trivial and itself involves many mechanistic assumptions and insights. While I generally appreciate the motivation of the study, I have concerns about some of the assumptions and feel that some aspects of the model behavior and study conclusions may be artifacts of these assumptions. Specifically, I feel that the key conclusions about speed-specificity tradeoffs and the time-course for attaining desired distributions are overstated. Below I indicate how other model assumptions, or even factoring another term into the same model (point 6), may overcome the stated limitations in achieving target cargo distributions.*

*1) The first aspect of the model involves the microtubule-based transport.*

*Here the authors adapt and simplify a biased 1-D random-walk model by*

*Muller et al. They consider two variants: simple independent probabilities,and history-dependent probabilities. This first set of model conversion iswell-grounded in the literature and results in a familiar model form. Thistakes them to an analysis of cargo distribution as a function of transportrates, and they show that a variety of distributions can be achieved throughthis mechanism. If I understand correctly, the main novelty in this figureis the result that even with long runs the mass-action model fits thestochastic one reasonably well.*

The reviewer understood the purpose of this figure correctly, but we have nonetheless clarified it in the revision.

*It would be nice to have a panel of this dependency, that is, a graph of fit vs. run length.*

We have added this as a supplemental figure (Figure 1—figure supplement 1).

*2) The next step of the modeling is to consider how the local rates oftransport might be modulated. The authors come up with a somewhat limitingmodel here, assuming only a single signal (Ca for convenience), and achieveforward/backward rate control by taking the signal levels in successive spatialcompartments. Here one could readily imagine that different localsignals might be a more versatile (and spatially more precise) way to achievecontrol of forward and backward rates of transport. Can the authors examinethis?*

There is likely to be a multitude of signals but the point is that these would somehow converge to have the net effectof controlling trafficking rates or detachment rates. One could assume an arbitrarily complex signaling pathway that achieves the same net result but for clarity we just illustrated how this might work for one signal, simply to show that it was feasible.

Moreover, we showed that even a single signal, such as calcium, can in principle allow arbitrarily precise trafficking. A more complex model involving multiple signals could not be more spatially precise, although it might work just as well.

*3) The electrical calculations in Figure 2 are poorly described. I assumethat the authors use the full Migliore 2012 model to obtain an electricalpotential distribution upon synaptic stimulation. It would have been usefulto have seen the electrical potential and its time-course. Over what timewas the 'average potential' taken? How did the potential map to the rates?*

*The authors refer to the methods section but there is insufficient informationthere.*

We have clarified this by completely re-writing this section (subsection “Biophysical formulation of the sushi belt model”, last paragraph and subsection “Simulations in realistic morphologies”, first paragraph) and simplified Figure 2 by removing extraneous panels. The reviewer nevertheless correctly understood the important technical points on his/her first read. We believe showing the electrical trace is not necessary as this would be a distraction from the main point, which is simply that local demand can vary and can be controlled by signals such as synaptic input. Details of the exact mapping between electrical signals and demand for cargo would not affect our results as these only depend on having a localized demand signal of some kind.

*4) The authors go through a few more elaborations in the model, before bringing in a 'detachment' scheme that finally takes their model to something more likethe full sushi-belt model. In Figure 1–Figure 3 I am concerned that the analysistalks about density of cargo on the motors rather than free cargo in thedendrites. First, it would be valuable to make this distinction clearer tothe reader.*

The reviewer arrived at the correct conclusions and interpretations. We have, however, completely re-written the paper to better motivate variants of the model according to thequestions these address. In particular, we present the full sushi-belt model up front in equation 2 and emphasize that Figure 1–Figure 3 refer to transport “on the motors.”

*Second, it would be valuable to discuss whether these predictionshave physiological observations to compare with. I do not have a sense forhow much cargo sits on the motors, and how much variability is observedin the distribution of motor-attached vs. detached forms.*

Despite an extensive literature search, we are also unaware of experimental studies that have given precise estimates of these parameters. We therefore examined a broad class of models over wide parameter ranges that should account for the unknown ratio of bound cargo (several fold). We have also added an entirely new analysis of a model with cargo recycling and excess cargo on the motors(subsection “Fine-tuned trafficking rates and cargo recycling introduce new tradeoffs”; subsection “Distinct cell-type morphologies face order of magnitude differences in speed, precision and efficiency of trafficking”, last paragraph; Figure 5—figure supplement 2 and Figure 6). Indeed, we hope our study will draw attention to these parameters and motivate experimentalists to make the appropriate measurements.

*5) When detachment is incorporated into the model, the authors find that onegets non-specific cargo delivery, as well as depletion of cargo attached tothe motors. I am concerned that these phenomena are more a reflection ofassumptions than physiology. Specifically, the unloading of cargo is anopen-loop, stimulus-driven process in the model. I wonder how many of thesefindings would hold if the unloading rate were driven not just by stimulus,but also by feedback based on amount of desired cargo that was alreadypresent. That is, a term dependent on*
ui**. Further, the degradation itselfcould also be driven by feedback. I suspect that the set point might bereached much faster with these elaborations.*

We assumed that some mechanism sets the trafficking rates to get a specific distribution of cargo. By assuming that the steady state is equal to the desired distribution we are implicitly assuming feedback. We have now made this clear by completely re-writing the manuscript (subsection “Biophysical formulation of the sushi belt model”, first paragraph). We have also now considered the effects of cargo reloading, which does indeed reduce the settling time.

All of our analysis assumes a simple mass-action model with fixed rates. This is the simplest non-trivial assumption. Arbitrarily allowing for any possible nonlinear feedback system would of course permit faster delivery. In the best case, if some mechanism could signal rapidly between neurites, the cell could shut off transport to dendrites that don't need cargo. We might further suppose that detachment rates approach zero when cargo is at target, and some very large (but finite) value otherwise. The lower bound on delivery time with these two assumptions is then the path length divided by the maximum velocity of the motors. This is the naive best-case transit time that we now discuss in the opening paragraph of the Discussion.

However, this best case is not particularly illuminating because, while physically possible, it requires us to invoke mechanisms that have never been shown to exist, and are not specified in the current, published versions of the sushi-belt model. Enumerating other kinds of mechanisms that can give intermediate performance is unlikely to be useful in the absence of empirical evidence.

On the other hand, the formulation of the sushi belt model we presented makes the simplest assumptions consistent with empirical data and the resulting predictions can be tested in a variety of ways. The consequences of the predictions and possible ways to test them are now discussed at length. As we emphasized in our opening, the value of our study is not confined to whether the predictions hold true, because a violation will tell us how to refine the current working model in the field.

*6) Looking in more detail at the equations in the subsection “Incorporating detachment and degradation into the mass-action model”, I was trying to understand the effect of loading density of the cargo. Specifically,if ui is large and the desired ui* is small, surely the system shouldgive a very rapid convergence to the target ui*? In other words, if thereis a huge amount of cargo available and going past, then one can quicklyobtain what one needs in any location to a high degree of accuracy. Itseems to me that the loading term should also play a role in the analysis on*

*Figure 5 and Figure 6. Thus the 'slow detachment' case could actually be fastin absolute time terms if one were to factor in lots of available cargo.*

*I do not see this factor in the analysis in the subsection “Conservative experimental estimates of trafficking parameters suggest that the tradeoff between speed and specificity is severe”.*

We agree. In the case of *reversible* detachment, adding large amounts of cargo can support fast and accurate transport in the model. We have added a new results and analyses addressing this point (subsection “Fine-tuned trafficking rates and cargo recycling introduce new tradeoffs”, subsection “Distinct cell-type morphologies face order of magnitude differences in speed, precision and efficiency of trafficking”, last paragraph; Figure 5—figure supplement 2 and Figure 6). However, to get very fast transport requires large cargo excesses which may be energetically inefficient to produce or may lead to “traffic jams” on the microtubules (Welte, 2004). We do not rule this out as a possibility, but present these results so that readers can judge for themselves.

We measure error in the relative concentration of cargo, that is, normalizing for the absolute amount. We do this because we did not want to focus on whether a neuron produces enough cargo overall, but instead on whether it distributes it appropriately. Moreover, relative abundances of signaling components are often what matters to tune spatial physiological properties (e.g. relative strengths of synaptic inputs).

*Reviewer #2:*

*In this manuscript, the authors developed a theoretical model for transport of cargoes on microtubules. Analytical solutions of the model show that such transport can either be fast or precise but not both. (Precision in this case means similarity to target cargo concentration at the destination.) In particular, the authors considered two different transport schemes were considered: (1) specific transport, uniform detachment, and (2) uniform transport, specific detachment.*

*1) A consequence of the first transport scheme is that bottlenecks will occur with the same probability in the main dendrites and the subsequent branch dendrites, since the rate constants within the whole neuron are modeled by the same function.*

The reviewer is correct that the same equation governs how trafficking rates are modulated by demand. However, the demand signal need not be uniform across the main dendrite and smaller branches. We think there may be a misunderstanding here, so we were careful to make the complete rewrite of this section clearer (subsection “Biophysical formulation of the sushi belt model”).

*However, it is reasonable to ask if neurons in reality do exhibit bottlenecks in the main dendrite and the branch/daughter dendrites at the same frequency.*

Our results simply state that if trafficking rates are controlled by local demand signals, then bottlenecks can occur. There are two things to draw from this: (1) ongoing physiological processes may actually cause bottlenecks and subsequent non-local effects in dendritic physiology; (2) experimental induction of bottlenecks provides a means to test whether trafficking is, in fact, controlled by local demand. We have completely rewritten the section on bottlenecks to clarify these points and their relevance (subsection “Transport bottlenecks occur when trafficking rates are non-uniform”). We agree it would be exciting to test this experimentally and we have offered suggestions as to how this can be done (see last paragraph of the aforementioned subsection).

*2) Perhaps there could be more detailed studies of transports using a combination of these two transport schemes (Figure 4). For example, will an intermediate strategy improve speed and precision, i.e., can a scheme involving intermediate transport-specificity and intermediate detachment-specificity circumvent the problems of bottlenecks and cargo leakage? Perhaps a phase space plot that illustrates the effect of the combinations of schemes on accuracy and transport time may help to convey the information better.*

We did consider this possibility and agree that it is important. In the new manuscript we draw the reader’s attention to Figure 5. In one scenario, the transport-specific model (which we now call DDT) outperforms the detachment-specific model (now DDD), while the opposite model performs better in the other scenario. Thus, changing the pattern of cargo demand will change which model performs best (including different intermediate models). In addition, Figure 5—figure supplement 1 also shows that strategies that are optimal for one pattern of demand fail to be optimal for others.

*Reviewer #3:*

*The manuscript by Williams applies a "sushi-belt delivery model" to cargo transport In CA1 pyramidal neurons. The goal is to understand the tradeoff between speed versus precision during cargo transport along microtubules by motor proteins. The manuscript opens with a rather general discussion (mass-action model) of convection-diffusion in a channel that is coarse-grained at the level of adjacent boxes. This idea is extended to model pyramidal neurons where the steady state distribution of cargo is calculated w.r.t. a target profile. The authors then model a bottleneck situation by assuming low cargo transition rates (epsilon) into a compartment, and test how the system converges to steady state as a function of epsilon. The model is taken further by introducing detachment from microtubules (possibly followed by diffusion-recapture – details unclear), and looking at efficiency of transport under two possibilities – cargo is selectively transported to target or is uniformly distributed (combined with detachment).*

*The goal is laudable because it attempts to present a generalized and simple mathematically solvable coarse-grained description of cargo localization in a complex neuronal geometry. Most of the assumptions of the mathematical model appear valid, their rate constants seem to match the experimental velocities and they seem to have taken into consideration various scenarios during cellular transport.*

We were encouraged by these general statements, particularly the acknowledgement that our assumptions appear valid and the recognition that the goal was to shed light on a complex biological system using a simple and interpretable model.

Given this, we were surprised by some of reviewer 3’s concerns which we address point- by-point below. We believe that many of these concerns were the result of the language and presentation of our original manuscript, which in retrospect was not very clearly presented. We have worked hard to improve the presentation and make the scope, purpose, predictions and caveats clearer.

*However, we feel that the paper starts off being rather general, and remains more-or-less so till the end.*

We do not agree that generality is a shortcoming in a scientific study. Moreover, we make numerous specific, testable predictions about global settling times of cargo in realistic neural morphologies, among other things. We have tried to make the more specific predictions and experimental questions clearer in the revised manuscript.

*For example, Figure 4 show that the target cargo distributions are always achieved irrespective of the transport/detachment ratio. What is one expected to learn from this, and how might it be useful to plan future experiments? If the message is that many strategies can be employed to achieve target distributions, then this is a rather weak message unless this theme is developed further with specific examples and suggestions.*

We think the reviewer may have misinterpreted the message of this figure, so we have completely rewritten the paper and introduced the family of trafficking strategies in a much more logical way. We show that there are many possible mechanisms, but crucially we show that they are not equivalent in testable ways. For example, we revealed transport bottlenecks as an experimentally measurable behavior that is particular to the DDT model (and are absent in DDD, as they are now called).

Most importantly, we show that this entire class of models is subject to a speed-accuracy tradeoff and that the optimal strategy depends on the spatial profile of demand (Figure 5). Thus, different neuron types with different morphologies might exhibit trafficking strategies that are tuned to better cope with, say proximal vs distal fluctuations in demand. We think that researchers who are interested in higher level regulation of trafficking will find this very relevant and not at all obvious.

*Similarly, the observation (subsection “Convergence rate”, last paragraph) that transport will achieve steady state faster if bottlenecks are removed – why is this surprising?*

What is surprising is that elevated global demand results in faster convergence than distal demand alone. One might naively expect the opposite. Furthermore, this is experimentally testable, as we discussed, yet no attempt has been made to observe this effect that we are aware of, which further supports the case that this is not an obvious prediction of the underlying trafficking model. Again, we were concerned that the writing in the original manuscript might have obscured this point. This and related sections have been completely rewritten.

*This part is followed up by a few poorly explained lines where the results (Figure 3) seem interesting, but are obscured by unnecessary usage of complicated Latin words.*

We have removed jargon, revised the figure and improved clarity on this result (subsection “Transport bottlenecks occur when trafficking rates are non-uniform”, Figure 3). This is an important point of our paper as it proposes a possible experimental approach for testing and characterizing the DDT model. Specifically, we propose that selectively stimulating different regions of the dendritic tree (as done in, e.g., Han & Heineman’s study) and tracking a cargo with activity-dependent trafficking (e.g., *Arc* mRNA) could expose the existence of bottlenecks (see last paragraph of the aforementioned subsection).

*On the same lines, in the places where it is mentioned, the connection to experiments is rather weak. Whether these assumptions hold true in a biological setting has not been tested for any neuronal cargoes. Live imaging to show that at least a few cargoes follow this model would have helped.*

We share the reviewer’s desire for detailed experimental studies that address these measurements. However, we believe this is an unreasonable request for this manuscript and in any case we do not have the facilities to perform these experiments. Secondly, all papers need a scope and ours is already substantial. As the reviewer initially appreciated, extracting meaningful, testable predictions from a conceptual model of a very complex process is extremely challenging. It is also crucial for progress because it is hard to identify compelling experiments – such as measurements of total settling times, or the bottleneck effects – without a rational analysis.

We would also like to point out that in this review process our paper has already triggered a passionate discussion of the need to measure quantities that nobody has previously measured, such as global settling times, based on the predictions of a current theoretical model. This is the kind of outcome we would anticipate from a successful theory paper.

*The authors talk about detachment and degradation of cargoes. But how does reloading of cargoes occur in instances when continuous supply is required?*

This is a very useful suggestion. As discussed elsewhere, we have examined this in the revised manuscript (new section on reloading, subsection “Fine-tuned trafficking rates and cargo recycling introduce new tradeoffs”, Figure 6 and Figure 5—figure supplement 2).

*The authors suggest using their model that accurate transport is slower, and faster rates of transport requires much greater complexity and is very sensitive to perturbations. This is hard to visualize in case of most neuronal functions where efficient robust and rapid signaling does occur during processes such as long term memory formation, signaling at the synapse etc. Again, there appears to be a disconnection between real biology and the model.*

We disagree with this conclusion. The reviewer is simply asserting that synaptic plasticity is fast and precise, and can be both fast and precise while relying on nucleus- to-synapse communication. We do not know of any published data directly supporting this assertion, and the reviewer does not cite any. Even in extensively studied systems like the *Aplysia* gill withdrawal circuit or the Shaffer collateral-CA1 synapse, the precision of synaptic plasticity is unknown. Similarly, there is little direct evidence that global trafficking is crucial for other forms of synaptic signaling on short timescales. In fact, there is mounting evidence, discussed and referenced at length in our manuscript, that local biosynthesis is important for many processes the reviewer alludes to. To understand why this is the case, we need to analyze the limitations of global transport, which is one of the goals of our paper.

*Taken together, we feel that this work would not have sufficient impact to warrant publication in eLife. This is in contrast to models of microtubule transport (e.g. Lipowsky group PNAS paper) which have made more specific mechanistic predictions that advanced the field and inspired new experiments.*

We appreciate the work reviewer appears to refer to (Klumpp and Lipowsky, 2005; Muller et al., 2008) and we cite these in our manuscript.

However, detailed models cannot address the high-level conceptual and physiological questions that are of interest to us and many other neurobiologists. Models that attempt to answer high-level questions using intricate mechanistic details are far more sensitive to assumptions and measurement error than appropriately coarse-grained models (see, for example, O'Leary, Sutton & Marder 2015, Curr Opinion in Neurobiology).

We feel it is a mistake to think that high-level, conceptual studies automatically lack 'impact' when the object of study is biochemical in nature. Conceptual questions are what motivate mechanistic work, after all.

Secondly, as the other reviewers appreciated, it is important to put mathematical flesh on the bones of the word-models that guide many experimentalists and other researchers who try to intuit the capabilities of global trafficking when formulating theories of how, say, synaptic plasticity might work.

Thirdly, general models can provide distinct and complementary insights to those gathered from detailed models and can make detailed studies more relevant to researchers outside a specific domain. For example, our study connects active transport to important questions in neuroanatomy and plasticity.

Finally, our study does make concrete predictions which, if addressed experimentally, would certainly advance the field. We return once again to the question of global settling times as an example: the reviewers expressed a strong belief that these would be faster than the model predicts, yet these have never been measured experimentally and there is no specific, existing motivation to do so. We believe our study can provide this motivation.

[Editors' note: the author responses to the re-review follow.]

*Essential revisions:*

*1) I'm surprised that the feedback and cargo recycling processes do not bringrapid settling to the system without large cargo excess. I think this is oneof the key findings of the paper. I would suggest that the authors move someof the panels from Figure 5—figure supplement 2 into the main body of the paper, so as to better present this result.*

We have moved the figures to the main results, as suggested. They now constitute Figure 6 and Figure 7 of the revised manuscript.

*2) The authors have based their entire simulation on a real life neuron (CA1 Pyramidal Cell) with a fixed number of compartments (742). It would be good if the authors throw some light on (i) how sensitive their simulations are to the number of compartments in the neuron, and (ii) how the compartment size is related to the cargo size.*

We have performed an additional analysis of the sensitivity of settling time to compartment size (new Figure 6—figure supplement 1) which is seen to asymptote as compartment number increases. Notably, coarsening (reducing number of compartments) decreases estimated settling time. Thus, spatial discretization makes our estimates of the severity of the speed-precision tradeoff conservative.